# The Contextual Lasso:
# Sparse Linear Models via Deep Neural Networks

**Ryan Thompson**[*]
University of New South Wales
CSIRO's Data61

**Amir Dezfouli**[†]
BIMLOGIQ

**Robert Kohn**
University of New South Wales

## Abstract

Sparse linear models are one of several core tools for interpretable machine learning, a field of emerging importance as predictive models permeate decision-making in many domains. Unfortunately, sparse linear models are far less flexible as functions of their input features than black-box models like deep neural networks. With this capability gap in mind, we study a not-uncommon situation where the input features dichotomize into two groups: explanatory features, which are candidates for inclusion as variables in an interpretable model, and contextual features, which select from the candidate variables and determine their effects. This dichotomy leads us to the contextual lasso, a new statistical estimator that fits a sparse linear model to the explanatory features such that the sparsity pattern and coefficients vary as a function of the contextual features. The fitting process learns this function nonparametrically via a deep neural network. To attain sparse coefficients, we train the network with a novel lasso regularizer in the form of a projection layer that maps the network's output onto the space of $\ell_1$-constrained linear models. An extensive suite of experiments on real and synthetic data suggests that the learned models, which remain highly transparent, can be sparser than the regular lasso without sacrificing the predictive power of a standard deep neural network.

## 1   Introduction

Sparse linear models—linear predictive functions in a small subset of features—have a long history in statistics, dating back at least to the 1960s (Garside, 1965). Nowadays, against the backdrop of elaborate, black-box models such as deep neural networks, the appeal of sparse linear models is largely their transparency and intelligibility. These qualities are sought in decision-making settings (e.g., consumer finance and criminal justice) and constitute the foundation of interpretable machine learning, a topic that has recently received significant attention (Murdoch et al., 2019; Molnar et al., 2020; Rudin et al., 2022; Marcinkevičs and Vogt, 2023). Interpretability, however, comes at a price when the underlying phenomenon cannot be predicted accurately without a more expressive model capable of well-approximating complex functions, such as a neural network. Unfortunately, one must forgo direct interpretation of expressive models and instead resort to post hoc explanations (Ribeiro et al., 2016; Lundberg and Lee, 2017), which have their own flaws (Laugel et al., 2019; Rudin, 2019).

Motivated by a desire for interpretability and expressivity, this paper focuses on a setting where sparse linear models and neural networks can collaborate together. The setting is characterized by a not-uncommon situation where the input features dichotomize into two groups, which we call explanatory features and contextual features. Explanatory features are features whose effects are of primary interest. They should be modeled via a low-complexity function such as a sparse linear model for interpretability. Meanwhile, contextual features describe the broader predictive context,

---

[*]Corresponding author. Email: `ryan.thompson1@unsw.edu.au`
[†]Part of this work was carried out while the author was at CSIRO's Data61.

37th Conference on Neural Information Processing Systems (NeurIPS 2023).

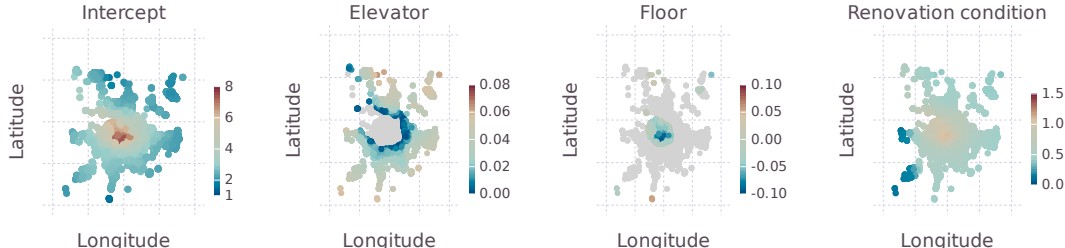

Figure 1: Fitted coefficient functions from the contextual lasso for the house pricing dataset. Colored points indicate coefficient values at different locations. Grey points indicate coefficients equal to zero.

e.g., the location of the prediction in time or space (see the house pricing example below). These inform which explanatory features are relevant and, for those that are, their exact low-complexity effects. Given this role, contextual features are best modeled via an expressive function class.

The explanatory-contextual feature dichotomy described above leads to the seemingly previously unstudied contextually sparse linear model:

$$g\left(\mathrm{E}[y \mid \mathbf{x}, \mathbf{z}]\right) = \sum_{j \in S(\mathbf{z})} x_j \beta_j(\mathbf{z}). \tag{1}$$

To parse the notation, $y \in \mathbb{R}$ is a response variable, $\mathbf{x} = (x_1, \ldots, x_p)^\top \in \mathbb{R}^p$ are explanatory features, $\mathbf{z} = (z_1, \ldots, z_m)^\top \in \mathbb{R}^m$ are contextual features, and $g$ is a link function (e.g., identity for regression or logit for classification).[3] Via the contextual features, the set-valued function $S(\mathbf{z})$ encodes the indices of the relevant explanatory features (typically, a small set of $j$s), while the coefficient functions $\beta_j(\mathbf{z})$ encode the effects of those relevant features. The model (1) draws inspiration from the varying-coefficient model (Hastie and Tibshirani, 1993; Fan and Zhang, 2008; Park et al., 2015), a special case that assumes all explanatory features are always relevant, i.e., $S(\mathbf{z}) = \{1, \ldots, p\}$ for all $\mathbf{z} \in \mathbb{R}^m$. We show that this new model is more powerful for various problems, including energy forecasting and disease prediction. For these tasks, sparsity patterns can be strongly context-dependent.

The main contribution of our paper is a new statistical estimator for (1) called the contextual lasso. The new estimator is inspired by the lasso (Tibshirani, 1996), a classic sparse learning tool with excellent properties (Hastie et al., 2015). We focus on tabular datasets as these are the most common use case for the lasso and its cousins. Whereas the lasso fits a sparse linear model that fixes the relevant features and their coefficients once and for all (i.e., $S(\mathbf{z})$ and $\beta_j(\mathbf{z})$ are constant), the contextual lasso fits a contextually sparse linear model that allows the relevant explanatory features and coefficients to change according to the prediction context. To learn the map from contextual feature vector to sparse coefficient vector, we use the expressive power of neural networks. Specifically, we train a feedforward neural network to output a vector of linear model coefficients sparsified via a novel lasso regularizer. In contrast to the lasso, which constraints the coefficients' $\ell_1$-norm, our regularizer constraints the *expectation* of the coefficients' $\ell_1$-norm with respect to $\mathbf{z}$. To implement this new regularizer, we include a novel projection layer at the bottom of the network that maps the network's output onto the space of $\ell_1$-constrained linear models by solving a convex optimization problem.

To briefly illustrate our proposal, we consider data on property sales in Beijing, China, studied in Zhou and Hooker (2022). We use the contextual lasso to learn a pricing model with longitude and latitude as contextual features. The response is price per square meter. Figure 1 plots the fitted coefficient functions of three property attributes (explanatory features) and an intercept. The relevance and effect of these attributes can vary greatly with location. The elevator indicator, e.g., is irrelevant throughout inner Beijing, where buildings tend to be older and typically do not have elevators. The absence of elevators also makes it difficult to access higher floors, hence the negative effect of floor on price. Beyond the inner city, the floor is mostly irrelevant. Naturally, renovations are valuable everywhere, but more so for older buildings in the inner city than elsewhere. The flexibility of the contextual lasso to add or remove attributes by location, and simultaneously determine their coefficients, equips sellers with personalized interpretable models containing only the attributes most relevant to them. At the same time, these models outpredict both the lasso and a deep neural network; see Appendix A.

---

[3]The intercept is omitted throughout this paper to ease notation.

The rest of paper is organized as follows. Section 2 introduces the contextual lasso and describes techniques for its computation. Section 3 discusses connections with earlier related work. Section 4 reports experiments on synthetic and real data. Section 5 closes the paper with a discussion.

## 2 Contextual lasso

This section describes our estimator. To facilitate exposition, we first rewrite the contextually sparse linear model (1) more concisely:

$$g\left(\mathrm{E}[y \,|\, \mathbf{x}, \mathbf{z}]\right) = \mathbf{x}^{\top} \boldsymbol{\beta}(\mathbf{z}).$$

The notation $\boldsymbol{\beta}(\mathbf{z}) := \left(\beta_1(\mathbf{z}), \ldots, \beta_p(\mathbf{z})\right)^{\top}$ represents a vector coefficient function which is sparse over its codomain. That is, for different values of $\mathbf{z}$, the output of $\boldsymbol{\beta}(\mathbf{z})$ contains zeros at different positions. The function $S(\mathbf{z})$, which encodes the set of active explanatory features in (1), is recoverable as $S(\mathbf{z}) := \{j : \beta_j(\mathbf{z}) \neq 0\}$.

### 2.1 Problem formulation

At the population level, the contextual lasso comprises a minimization of the expectation of a loss function subject to an inequality on the expectation of a constraint function:

$$\min_{\boldsymbol{\beta} \in \mathcal{F}} \quad \mathrm{E}\left[l\left(\mathbf{x}^{\top} \boldsymbol{\beta}(\mathbf{z}), y\right)\right] \qquad \mathrm{s.\,t.} \quad \mathrm{E}\left[\|\boldsymbol{\beta}(\mathbf{z})\|_1\right] \leq \lambda, \tag{2}$$

where the set $\mathcal{F}$ is a class of functions that constitute feasible solutions and $l : \mathbb{R}^2 \to \mathbb{R}$ is the loss function, e.g., square loss $l(\hat{y}, y) = (y - \hat{y})^2$ for regression or logistic loss $l(\hat{y}, y) = -y \log(\hat{y}) - (1 - y) \log(1 - \hat{y})$ for classification. Here, the expectations are taken with respect to the random variables $y$, $\mathbf{x}$, and $\mathbf{z}$. The parameter $\lambda \geq 0$ controls the level of regularization. Smaller values of $\lambda$ encourage $\boldsymbol{\beta}(\mathbf{z})$ towards zero over more of its codomain. Larger values have the opposite effect. The contextual lasso thus generalizes the lasso, which learns $\boldsymbol{\beta}(\mathbf{z})$ as a constant:

$$\min_{\boldsymbol{\beta}} \quad \mathrm{E}\left[l\left(\mathbf{x}^{\top} \boldsymbol{\beta}, y\right)\right] \qquad \mathrm{s.\,t.} \quad \|\boldsymbol{\beta}\|_1 \leq \lambda.$$

To reiterate the difference: the lasso coaxes the *fixed coefficients* $\boldsymbol{\beta}$ towards zero, while the contextual lasso coaxes the *expectation of the function* $\boldsymbol{\beta}(\mathbf{z})$ to zero. The result for the latter is coefficients that can change in value and sparsity with $\mathbf{z}$, provided the function class $\mathcal{F}$ is suitably chosen.

Given a sample $(y_i, \mathbf{x}_i, \mathbf{z}_i)_{i=1}^n$, the data version of the population problem (2) replaces the unknown expectations with their sample counterparts:

$$\min_{\boldsymbol{\beta} \in \mathcal{F}} \quad \frac{1}{n} \sum_{i=1}^{n} l\left(\mathbf{x}_i^{\top} \boldsymbol{\beta}(\mathbf{z}_i), y_i\right) \qquad \mathrm{s.\,t.} \quad \frac{1}{n} \sum_{i=1}^{n} \|\boldsymbol{\beta}(\mathbf{z}_i)\|_1 \leq \lambda. \tag{3}$$

The set of feasible solutions to optimization problem (3) are coefficient functions that lie in the $\ell_1$-ball of radius $\lambda$ when averaged over the observed data.[4] To operationalize this estimator, we take $\mathcal{F} = \{\boldsymbol{\beta}_{\mathbf{w}}(\mathbf{z}) : \mathbf{w}\}$, where $\boldsymbol{\beta}_{\mathbf{w}}(\mathbf{z})$ is a certain architecture of feedforward neural network (described shortly) parameterized by weights $\mathbf{w}$. This choice leads to our core proposal:

$$\min_{\mathbf{w}} \quad \frac{1}{n} \sum_{i=1}^{n} l\left(\mathbf{x}_i^{\top} \boldsymbol{\beta}_{\mathbf{w}}(\mathbf{z}_i), y_i\right) \qquad \mathrm{s.\,t.} \quad \frac{1}{n} \sum_{i=1}^{n} \|\boldsymbol{\beta}_{\mathbf{w}}(\mathbf{z}_i)\|_1 \leq \lambda. \tag{4}$$

Configuring a feedforward neural network such that its outputs are sparse and satisfy the $\ell_1$-constraint is not trivial. We introduce a novel network architecture to address this challenge.

### 2.2 Network architecture

The neural network architecture—depicted in Figure 2—involves two key components. The first and most straightforward component is a vanilla feedforward network $\boldsymbol{\eta}(\mathbf{z}) : \mathbb{R}^m \to \mathbb{R}^p$. The purpose of the network is to capture the nonlinear effects of the contextual features on the explanatory features.

---

[4]The $\ell_1$-ball is the convex compact set $\{\mathbf{x} \in \mathbb{R}^p : \|\mathbf{x}\|_1 \leq \lambda\}$.

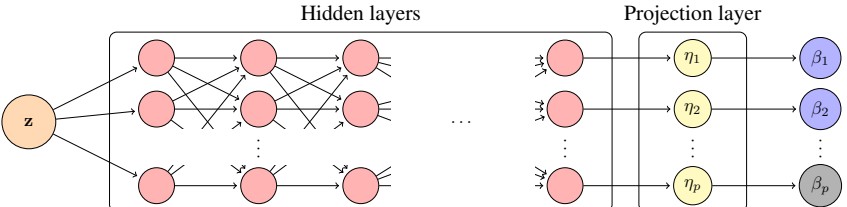

Hidden layers       Projection layer

Figure 2: Network architecture. The contextual features $\mathbf{z}$ pass through a series of hidden layers. The resulting dense coefficients $\eta_1, \ldots, \eta_p$ then enter a projection layer to produce sparse coefficients $\beta_1, \ldots, \beta_p$. Here, the last coefficient is gray to illustrate that it is zeroed-out by the projection layer.

Since the network involves only hidden layers with standard affine transformations and nonlinear maps (e.g., rectified linear activation functions), the coefficients they produce generally do not satisfy the contextual lasso constraint and are not sparse. To enforce the constraint and attain sparsity, we employ a novel projection layer as the second main component of our network architecture.

The projection layer takes the dense coefficients $\boldsymbol{\eta}(\mathbf{z})$ from the network and maps them to sparse coefficients $\boldsymbol{\beta}(\mathbf{z})$ by performing a projection onto the $\ell_1$-ball. Because the contextual lasso does not constrain each coefficient vector to the $\ell_1$-ball, but rather constrains the *average* coefficient vector, we project all $n$ coefficient vectors $\boldsymbol{\eta}(\mathbf{z}_1), \ldots, \boldsymbol{\eta}(\mathbf{z}_n)$ together. That is, we take the final sparse coefficients $\boldsymbol{\beta}(\mathbf{z}_1), \ldots, \boldsymbol{\beta}(\mathbf{z}_n)$ as the minimizing arguments of a convex optimization problem:

$$\boldsymbol{\beta}(\mathbf{z}_1), \ldots, \boldsymbol{\beta}(\mathbf{z}_n) := \underset{\boldsymbol{\beta}_1, \ldots, \boldsymbol{\beta}_n : \frac{1}{n}\sum_{i=1}^{n} \|\boldsymbol{\beta}_i\|_1 \leq \lambda}{\arg\min} \frac{1}{n}\sum_{i=1}^{n} \|\boldsymbol{\eta}(\mathbf{z}_i) - \boldsymbol{\beta}_i\|_2^2. \tag{5}$$

The minimizers of this optimization problem are typically sparse thanks to the geometry of the $\ell_1$-ball. The idea of including optimization as a layer in a neural network is explored in previous works (see, e.g., Amos and Kolter, 2017; Agrawal et al., 2019). Yet, to our knowledge, no previous work has studied optimization layers (also known as implicit layers) for inducing sparsity in a neural network.

The optimization problem (5) does not admit an analytical solution, though it is solvable by general purpose convex solvers (see, e.g., Boyd and Vandenberghe, 2004). However, because (5) is a highly structured problem, it is also amenable to more specialized algorithms. Such algorithms facilitate the type of scalable computation necessary for deep learning. Duchi et al. (2008) provide a low-complexity algorithm for solving (5) when $n = 1$. Algorithm 1 below is an extension to $n \geq 1$. The algorithm consists of two main steps: (1) computing a thresholding parameter $\theta$ and (2) soft-

---

**Algorithm 1** Projection onto $\ell_1$-ball

---

**input** Dense coefficients $\boldsymbol{\eta}_1, \ldots, \boldsymbol{\eta}_n$ and radius $\lambda$
Set $\boldsymbol{\mu} = (|\boldsymbol{\eta}_1^\top|, \ldots, |\boldsymbol{\eta}_n^\top|)^\top$
Sort $\boldsymbol{\mu}$ in decreasing order: $\mu_i \geq \mu_j$ for all $i < j$
Set $k_{\max} = \max\left\{k : \mu_k > \left(\sum_{l=1}^{k} \mu_l - n\lambda\right)/k\right\}$
Set $\theta = \left(\sum_{k=1}^{k_{\max}} \mu_k - n\lambda\right)/k_{\max}$
Compute $\boldsymbol{\beta}_1, \ldots, \boldsymbol{\beta}_n$ as $\beta_{ij} = \text{sign}(\eta_{ij})\max(|\eta_{ij}| - \theta, 0)$ for $i = 1, \ldots, n$ and $j = 1, \ldots, p$
**output** Sparse coefficients $\boldsymbol{\beta}_1, \ldots, \boldsymbol{\beta}_n$

---

thresholding the inputs using the computed $\theta$. Critically, the operations comprising Algorithm 1 are suitable for computation on a GPU, meaning the model can be trained end-to-end at scale.

Although the projection algorithm involves some nondifferentiable operations, most deep learning libraries provide gradients for these operations, e.g., sort is differentiated by permuting the gradients and abs is differentiated by taking the subgradient zero at zero. The gradient obtained by automatically differentiating through Algorithm 1 is the same as that from implicitly differentiating through a convex solver (e.g., using `cvxpylayers` of Agrawal et al., 2019), though the latter is slower. In subsequent work, Thompson et al. (2023) derive analytical gradients for a projection layer that maps matrices onto an $\ell_1$-ball. Their result readily adapts to the vector case dealt with here.

Computation of the thresholding parameter is performed only during training. For inference, the estimate $\hat{\theta}$ from the training set is used for soft-thresholding. That is, rather than using Algorithm 1

as an activation function when performing inference, we use $T(x) := \text{sign}(x) \max(|x| - \hat{\theta}, 0)$. The purpose of using the estimate $\hat{\theta}$ rather than recomputing $\theta$ via the algorithm is because the $\ell_1$-constraint applies to the *expected* coefficient vector. It need not be the case that every coefficient vector produced at inference time lies in the $\ell_1$-ball, which would occur if the algorithm is rerun.

Algorithm 1 computes the thresholding parameter using the full batch of $n$ observations. The algorithm can also be applied to mini-batches during training. Once training is complete, the estimate $\hat{\theta}$ for inference can be obtained via a single forward pass of the full batch through the network.

### 2.3 Grouped explanatory features

In certain settings, the explanatory features may be organized into groups such that all the features in a group should be selected together. These groups may emerge naturally (e.g., genes in the same biological path) or be constructed for a statistical task (e.g., basis expansions for nonparametric regression). The prevalence of such problems has led to the development of sparse estimators capable of handling groups, one of the most well-known being the group lasso (Yuan and Lin, 2006; Meier et al., 2008). Perhaps unsurprisingly, the contextual lasso extends gracefully to grouped selection.

Let $\mathcal{G}_1, \ldots, \mathcal{G}_g \subseteq \{1, \ldots, p\}$ be a set of $g$ nonoverlapping groups, and let $\boldsymbol{\beta}_k(\mathbf{z})$ and $\mathbf{x}_k$ be the coefficient function and explanatory features restricted to group $\mathcal{G}_k$. In the noncontextual setting, the group lasso replaces the $\ell_1$-norm $\|\boldsymbol{\beta}\|_1$ of the lasso with a sum of group-wise $\ell_2$-norms $\sum_{k=1}^{g} \|\boldsymbol{\beta}_k\|_2$. To define the *contextual group lasso*, we make the analogous modification to (4):

$$\min_{\mathbf{w}} \quad \frac{1}{n} \sum_{i=1}^{n} l \left( \sum_{k=1}^{g} \mathbf{x}_{ik}^{\top} \boldsymbol{\beta}_{k,\mathbf{w}}(\mathbf{z}_i), y_i \right) \qquad \text{s.t.} \quad \frac{1}{n} \sum_{i=1}^{n} \sum_{k=1}^{g} \|\boldsymbol{\beta}_{k,\mathbf{w}}(\mathbf{z}_i)\|_2 \leq \lambda.$$

Similar to how the absolute values of the $\ell_1$-norm are nondifferentiable at zero, which causes individual explanatory features to be selected, the $\ell_2$-norm is nondifferentiable at the zero vector, causing grouped explanatory features to be selected together. To realize the grouped estimator, we adopt the same architecture as before but replace the previous (ungrouped) projection layer with its grouped counterpart. This change demands a different projection algorithm, presented in Appendix B.

### 2.4 Side constraints

Besides the contextual (group) lasso constraint, our architecture readily accommodates side constraints on $\boldsymbol{\beta}(\mathbf{z})$ via modifications to the projection. For instance, we follow Zhou and Hooker (2022) in the house pricing example (Figure 1) and constrain the coefficients on the elevator and renovation features to be nonnegative. Such sign constraints reflect domain knowledge that these features should not impact price negatively. Appendix C presents the details and proofs of this extension.

### 2.5 Pathwise optimization

The lasso regularization parameter $\lambda$, controlling the size of the $\ell_1$-ball and thus the sparsity level, is typically treated as a tuning parameter. For this reason, algorithms for the lasso usually provide multiple models over a grid of varying $\lambda$, which can then be compared (Friedman et al., 2010). Towards this end, it can be computationally efficient to compute the models pathwise by sequentially warm-starting the optimizer. As Friedman et al. (2007) point out, pathwise computation for many $\lambda$ can be as fast as for a single $\lambda$. For the contextual lasso, warm starts also reduce run time compared with initializing at random weights. More importantly, however, pathwise optimization improves the training quality. This last advantage is a consequence of the network's nonconvex optimization surface. Building up a sophisticated network from a simple one helps the optimizer navigate this surface. Appendix D presents our pathwise algorithm and an approach for setting the $\lambda$ grid.

### 2.6 Relaxed fit

A possible drawback to the contextual lasso, and indeed all lasso estimators, is bias of the linear model coefficients towards zero. This bias, which is a consequence of shrinkage from the $\ell_1$-norm, can help or hinder depending on the data. Typically, bias is beneficial when the number of observations is low or the level of noise is high, while the opposite is true in the converse situation (see, e.g., Hastie et al., 2020). This consideration motivates a relaxation of the contextual lasso that unwinds some, or all, of

the bias imparted by the $\ell_1$-norm. We describe an approach in Appendix E that extends the proposal of Hastie et al. (2020) for relaxing the lasso. Their relaxation, which simplifies an earlier proposal by Meinshausen (2007), involves a convex combination of the lasso's coefficients and "polished" coefficients from an unregularized least squares fit on the lasso's selected features. We extend this idea from the lasso's fixed coefficients to the contextual lasso's varying coefficients. The benefits of the relaxation are demonstrated empirically in Appendix E, where we present an ablation study.

## 2.7 Computational complexity

A forward or backward pass through the vanilla feedforward component of the network takes $O(md + hd^2 + pd)$ time, where $h$ is the number of hidden layers, $d$ is the number of nodes per layer, $m$ is the number of contextual features, and $p$ is the number of explanatory features. A forward or backward pass through the projection algorithm takes $O(p)$ time (Duchi et al., 2008). The time complexity for a pass through the full network over $n$ observations is thus $O(nd(m + hd + p))$. This result suggests that the training time is linear in the sample size $n$ and number of features $m$ and $p$. Actual training times are reported in Appendix F that demonstrate linear complexity empirically.

## 2.8 Package

We implement the contextual lasso as described in this section in the `Julia` (Bezanson et al., 2017) package `ContextualLasso`. For training the neural network, we use the deep learning library `Flux` (Innes et al., 2018). Though the experiments throughout this paper involve square or logistic loss functions, our package supports *any* differentiable loss function, e.g., those in the family of generalized linear models (Nelder and Wedderburn, 1972). `ContextualLasso` is available at

https://github.com/ryan-thompson/ContextualLasso.jl.

# 3 Related work

The contextual explanation networks in Al-Shedivat et al. (2020) are cousins of the contextual lasso. These neural networks input contextual features and output interpretable models for explanatory features. They include the (nonsparse) contextual linear model, a special case of the contextual lasso where $\lambda = \infty$. In their terminology, the contextual linear model is a "linear explanation" model with a "deterministic encoding" function. They also explore a "constrained deterministic encoding" function that involves a weighted combination of individual fixed linear models with weights determined by the contextual features. To avoid overfitting, they apply $\ell_1$-regularization to the individual models. However, they have no mechanism that encourages the network to combine these sparse models such that the result is sparse. In contrast, the contextual lasso directly regularizes the sparsity of its outputs.

The contextual lasso is also related to several estimators that allow varying sparsity patterns. Yamada et al. (2017) devise the first of these—the localized lasso—which fits a linear model with a different coefficient vector for each observation. The coefficients are sparsified using a lasso regularizer that relies on the availability of graph information to link the observations. Yang et al. (2022) and Yoshikawa and Iwata (2022) follow with LLSPIN and NGSLL, neural networks that produce linear models with varying sparsity patterns via gating mechanisms. These approaches are distinct from our own, however. First, they do not dichotomize into $\mathbf{x}$ and $\mathbf{z}$, making the resulting model $\mathbf{x}^\top \boldsymbol{\beta}(\mathbf{x})$ difficult to interpret. Second, the sparsity level (NGSSL) or nonzero coefficients (LLSPIN) are fixed across observations, making them unsuitable for the contextual setting where both may vary.

Hallac et al. (2015) introduce the network lasso which has a different coefficient vector per observation, clustered using a lasso-style regularizer. They consider problems similar to ours, for which contextual information is available, but do not impose sparsity on the coefficients. Deleu and Bengio (2021) induce structured sparsity over neural network weights to obtain smaller, pruned networks that admit efficient computation. In our work, we leave the weights as dense and instead induce sparsity over the network's output for interpretability. Wang et al. (2020) propose a network quantization scheme with activation functions that output zeros and ones. Though our approach involves an activation that outputs zeros, we also allow a continuous output. Moreover, their end goal differs from ours; whereas they pursue sparsity to reduce computational complexity, we pursue sparsity for interpretability.

Our work also advances the broader literature at the intersection of feature sparsity and neural networks, an area that has gained momentum over the last few years. See, e.g., the lassonet of Lemhadri et al. (2021a,b) which selects features in a residual neural network using an $\ell_1$-regularizer on the skip connection. This regularizer is combined with constraints that force a feature's weights on the first hidden layer to zero whenever its skip connection is zero. See also Scardapane et al. (2017) and Feng and Simon (2019) for earlier ideas based on the group lasso, and Chen et al. (2021) for another approach. Though related, these methods differ from the contextual lasso in that they involve uninterpretable neural networks with fixed sparsity patterns. The underlying optimization problems also differ—whereas these methods regularize the network's weights, ours regularizes its outputs.

## 4 Experiments

The contextual lasso is evaluated here via experimentation on synthetic and real data. As benchmark methods, we consider the (nonsparse) contextual linear model, which uses no projection layer, and a deep neural network, which receives all explanatory and contextual features as inputs.[5] We further include the lasso, lassonet, and LLSPIN, which also receive all features. The localized lasso does not scale to the experiments that follow, so we instead compare it with the contextual lasso on smaller experiments in Appendix G. Appendix H provides the implementation details of all methods.

### 4.1 Synthetic data

We consider three different settings of increasing complexity: (1) $p = 10$ and $m = 2$, (2) $p = 50$ and $m = 2$, and (3) $p = 50$ and $m = 5$. Within each setting, the sample size ranges from $n = 10^2$ to $n = 10^5$. The full simulation design is detailed in Appendix I. As a prediction metric, we report the square or logistic loss relative to the intercept-only model. As an interpretability metric, we report the proportion of nonzero features. As a selection metric, we report the F1-score of the selected features; a value of one indicates all true positives recovered and no false positives.[6] All three metrics are evaluated on a testing set with tuning on a validation set, both constructed by drawing $n$ observations independently of the training set. Figure 3 reports the results for regression (i.e., continuous response).

The contextual lasso performs comparably with most of its competitors when the sample size is small. On the other hand, the contextual linear model (the contextual lasso's unregularized counterpart) can perform poorly here (its relative loss had to be omitted from some plots to maintain the aspect ratio). As $n$ increases, the contextual lasso begins to outperform other methods in prediction, interpretability, and selection. Eventually, it learns the correct map from contextual features to relevant explanatory features, recovering only the true nonzeros. Though its unregularized counterpart performs nearly as well in terms of prediction for large $n$, it remains much less interpretable, using all explanatory features. In contrast, the contextual lasso uses just 10% of the explanatory features on average.

The deep neural network's performance is underwhelming for most $n$. Only for large sample sizes does it begin to approach the prediction performance of the contextual lasso. The lassonet often performs somewhere between the two. These three methods should predict equally well for large enough $n$, though the function learned by the deep neural network and lassonet will remain opaque. The lasso makes some gains with increasing sample size, but lacks the expressive power of the contextual lasso needed to adapt to the complex sparsity pattern of the true model. LLSPIN—the only other method to allow for varying sparsity patterns—is the second best feature selector for $p = 10$, though its mediocre performance more generally is likely due to it not exploiting the explanatory-contextual feature dichotomy and not allowing its nonzero coefficients to change.

### 4.2 Energy consumption data

We consider a real dataset containing energy readings for a home in Mons, Belgium (Candanedo et al., 2017). Besides this continuous response feature, the dataset also contains $p = 25$ explanatory features in the form of temperature and humidity readings in different rooms of the house and local weather

---

[5]The contextual linear model corresponds to the contextual explanation network with a linear explanation and deterministic encoding in Al-Shedivat et al. (2020).

[6]The F1-score $:= 2\,\mathrm{TP}\,/(2\,\mathrm{TP} + \mathrm{FP} + \mathrm{FN})$, where TP, FP, and FN are the number of true positive, false positive, and false negative selections.

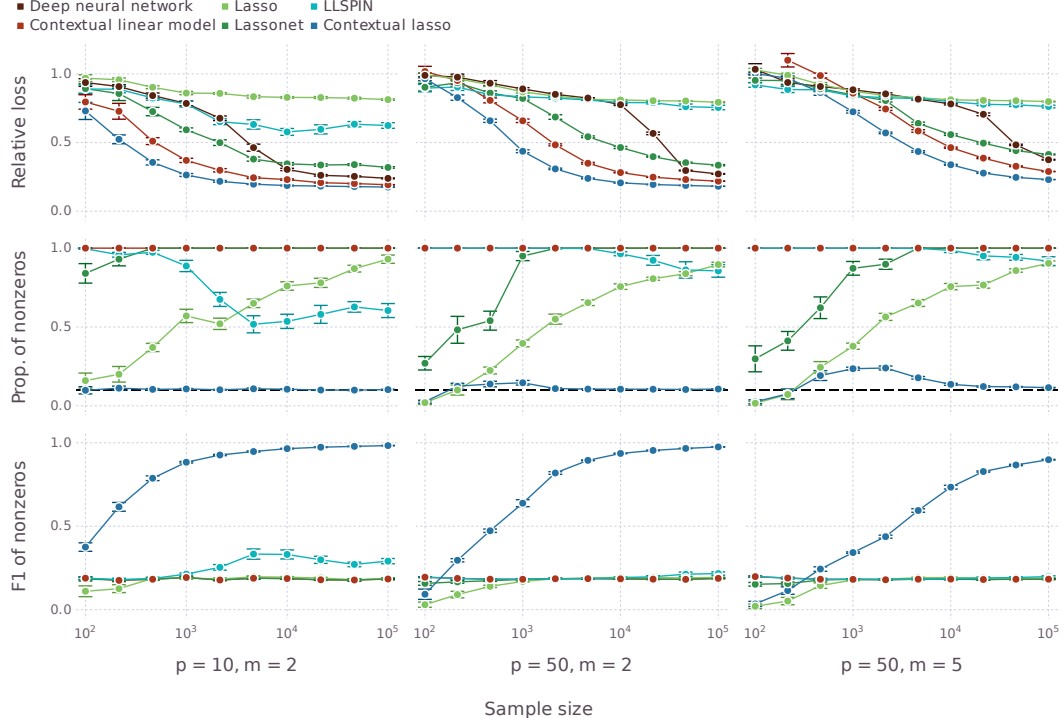

Figure 3: Comparisons on synthetic regression data. Metrics are aggregated over 10 synthetic datasets. Solid points are averages and error bars are standard errors. Dashed horizontal lines in the middle row indicate the true sparsity level.

data. We define several contextual features from the time stamp to capture seasonality: month of year, day of week, hour of day, and a weekend indicator. To reflect their cyclical nature, the first three contextual features are transformed using a sine and cosine, leading to $m = 7$ contextual features.

The dataset, containing $n = 19,375$ observations, is randomly split into training, validation, and testing sets in 0.6-0.2-0.2 proportions. We repeat this random split 10 times, each time recording performance on the testing set, and report the aggregate results in Table 1. As performance metrics, we consider the relative loss and average sparsity level (i.e., average number of selected explanatory features). Among all methods, the contextual lasso leads to the lowest test loss, outperforming even

Table 1: Comparisons on the energy consumption data. Metrics are aggregated over 10 random splits of the data. Averages and standard errors are reported.

|  | Relative loss | Avg. sparsity |
| --- | --- | --- |
| Deep neural network | $0.433 \pm 0.004$ | $25.0 \pm 0.0$ |
| Contextual linear model | $0.387 \pm 0.003$ | $25.0 \pm 0.0$ |
| Lasso | $0.690 \pm 0.002$ | $11.6 \pm 0.4$ |
| Lassonet | $0.423 \pm 0.003$ | $25.0 \pm 0.0$ |
| LLSPIN | $0.639 \pm 0.005$ | $24.5 \pm 0.1$ |
| Contextual lasso | $0.356 \pm 0.003$ | $2.8 \pm 0.4$ |

the deep neural network and lassonet.[7] Importantly, this excellent prediction performance is achieved while maintaining a high level of interpretability. In contrast to most other methods, which use all (or nearly all) available explanatory features, the predictions from the contextual lasso arise from linear models containing just 2.8 explanatory features on average! These linear models are also much simpler than those from the lasso, which typically involve more than four times as many features.

---

[7]The lassonet with tuned $\lambda$ uses nearly all features here. However, manually choosing $\lambda$ to attain a sparsity level similar to the contextual lasso substantially degrades its performance.

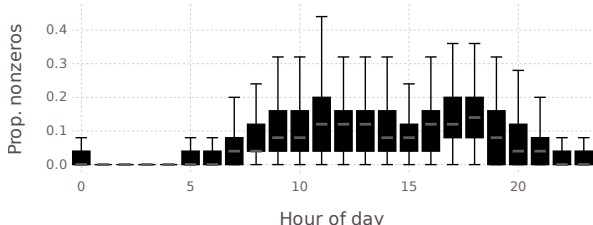

Figure 4: Explanatory feature sparsity as a function of hour of day for the estimated energy consumption model. The sparsity level varies within each hour because the other contextual features vary.

The good predictive performance of the contextual lasso suggests a seasonal pattern of sparsity. To investigate this phenomenon, we apply the fitted model to a randomly sampled testing set and plot the resulting sparsity levels as a function of the hour of day in Figure 4. The model is typically highly sparse in the late evening and early morning. Between 10 pm and 6 am, the median proportion of nonzero coefficients is 0%. There is likely little or no activity inside the house at these times, so sensor readings from within the house—which constitute the majority of the explanatory features—are irrelevant. The number of active explanatory features rises later in the day, peaking around lunchtime and dinnertime. Overall, a major benefit of the contextual lasso, besides its good predictions, is the ability to identify a parsimonious set of factors driving energy use at any given time of day.

### 4.3 Parkinson's telemonitoring data

We illustrate the contextual lasso on *grouped explanatory features* using data from a study on the progression of Parkinson's disease in 42 patients (Tsanas et al., 2009). The task is to predict disease progression (a continuous variable) using 16 vocal characteristics of the patients as measured at different times throughout the study. As Tsanas et al. (2009) point out, these vocal characteristics can relate nonlinearly to disease progression. To account for these effects, we compute a five-term cubic regression spline per explanatory feature ($p = 16 \times 5 = 80$). Each spline forms a single group of explanatory features ($g = 16$). The contextual features are the age and sex of the patients ($m = 2$).

The dataset of $n = 5,875$ observations is again partitioned into training, validation, and testing sets in the same proportions as before. As a new benchmark, we evaluate the group lasso, which is applied to splines of all explanatory and contextual features. The deep neural network and lassonet are applied to the original (nonspline) features.[8] The lasso is also applied to the original features to serve as a linear benchmark. The results are reported in Table 2. The purely linear estimator—the lasso–performs

Table 2: Comparisons on the Parkinson's telemonitoring data. Metrics are aggregated over 10 random splits of the data. Averages and standard errors are reported.

|                        | Relative loss       | Avg. sparsity    |
| ---------------------- | ------------------- | ---------------- |
| Deep neural network    | $0.367 \pm 0.015$   | $16.0 \pm 0.0$   |
| Lasso                  | $0.885 \pm 0.005$   | $3.1 \pm 0.1$    |
| Group lasso            | $0.710 \pm 0.006$   | $4.2 \pm 0.4$    |
| Lassonet               | $0.263 \pm 0.007$   | $15.5 \pm 0.2$   |
| Contextual group lasso | $0.113 \pm 0.006$   | $1.6 \pm 0.3$    |

worst overall. The group lasso improves over the lasso, supporting claims of nonlinearity in the data. The contextual group lasso is, however, the star of the show. Its models predict nearly three-times better than the next best competitor (lassonet) and are sparser than those from any other method.

Setting aside predictive accuracy, a major benefit of the contextual group lasso (compared with the deep neural network and lassonet) is that it remains highly interpretable. To illustrate, we consider the fitted spline function (i.e., the spline multiplied by its coefficients from the contextual group lasso) on the detrended fluctuation analysis (DFA) feature, which characterizes turbulent noise in speech. Figure 5 plots the function at three different ages of patient. For 70-year-olds, the function is zero, indicating DFA is not yet a good predictor of Parkinson's. At 75, the function becomes nonzero,

---

[8]Inputting the splines to these methods does not improve their performance.

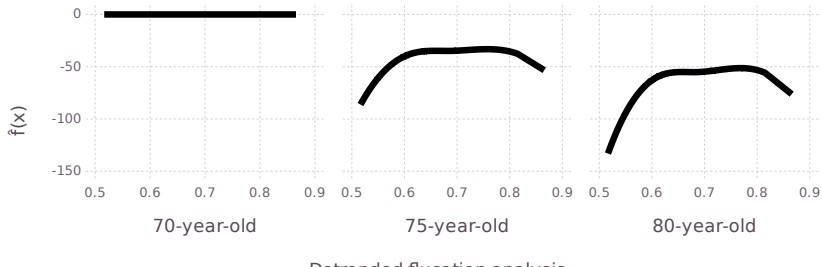

Figure 5: Fitted spline function from the contextual lasso for the detrended fluctuation analysis (DFA) explanatory feature. The age explanatory feature is varied while the sex feature is set to female.

taking on a concave shape. It becomes even more concave and negative 80. The models reported in Tsanas et al. (2009) also had this coefficient negative, but fixed across all ages. In contrast, the contextual lasso identifies DFA and other features as relevant only for patients of certain ages and sex.

### 4.4 Additional experiments

Experiments for classification on synthetic and real data are available in Appendix J. In Appendix K, we report high-dimensional experiments with $p = 1,000$ and fixed coefficient experiments (i.e., the lasso's home court). Appendix L investigates the stability of the contextual lasso with respect to the random initialization. Appendix M provides hyperlinks to the datasets used throughout the paper.

## 5 Discussion

Contextual sparsity is an important extension of the classical notion of feature sparsity. Rather than fix the relevant features once and for all, contextual sparsity allows feature relevance to depend on the prediction context. To tackle this intricate statistical learning problem, we devise the contextual lasso. This new estimator utilizes the expressive power of deep neural networks to learn interpretable sparse linear models with sparsity patterns that vary with the contextual features. The optimization problem that defines the contextual lasso is solvable at scale using modern deep learning frameworks. Grouped explanatory features and side constraints are readily accommodated by the contextual lasso's neural network architecture. An extensive experimental analysis of the new estimator illustrates its good prediction, interpretation, and selection properties in various settings. To the best of our knowledge, the contextual lasso is the only tool currently available for handling the contextually sparse setting.

The problem of deciding the explanatory-contextual feature split is the same as that faced with varying-coefficient models. Though the literature on varying-coefficient models is extensive, there are no definitive rules for partitioning the features in general. In the housing and energy examples, the contextual features are spatial or temporal effects, which are distinct from the remaining features. In the telemonitoring example, the patient attributes (age and sex) differ fundamentally from the vocal characteristics. Ultimately, the partition for any given application should be guided by domain expertise with consideration to the end goal. If one needs to interpret the exact effect of a feature, that feature should be an explanatory feature. If a feature's effect is of secondary interest, or it is suspected that the feature influences the structural form of the model, that feature should be a contextual feature. If the user determines there are no contextual features, the ordinary lasso is a more appropriate tool.

It remains an important avenue of future research to establish a solid theoretical foundation for the contextual lasso. The statistical properties of the lasso in terms of estimation, prediction, and selection are now well-established in theory (Bunea et al., 2007; Raskutti et al., 2011; Shen et al., 2013; Zhang et al., 2014). The synthetic experiments in our paper suggest that the contextual lasso satisfies similar properties, though theoretically establishing these results is challenging. Statistical convergence results for vanilla feedforward neural networks (e.g., Schmidt-Hieber, 2020) do not apply in our setting due to the projection layer. Moreover, to our knowledge, no statistical guarantees exist for neural networks configured with convex optimization layers that otherwise might apply here. It is also important to understand when the contextual lasso's performance is matched by a deep neural network, since both should predict well for large samples in the contextually sparse linear regime.

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

## A  House pricing data

We expand the analysis in Section 1 and compare the statistical performance of the contextual lasso with its competitors using the house pricing data. Following Zhou and Hooker (2022), the explanatory features are the elevator indicator, renovation condition, floor of the apartment, and number of living rooms and bathrooms ($p = 5$). The contextual features are longitude and latitude ($m = 2$). We randomly sample training, validation, and testing sets of size $n = 15,000$ and report the results across 10 random data splits in Table 3. The contextual lasso delivers sparse models and with competitive

Table 3: Comparisons of methods on the house pricing data. Metrics are aggregated over 10 random splits of the data. Averages and standard errors are reported.

|  | Relative loss | Avg. sparsity |
|---|---|---|
| Deep neural network | $0.515 \pm 0.003$ | $5.0 \pm 0.0$ |
| Contextual linear model | $0.505 \pm 0.002$ | $5.0 \pm 0.0$ |
| Lasso | $0.892 \pm 0.001$ | $5.0 \pm 0.0$ |
| Lassonet | $0.521 \pm 0.003$ | $5.0 \pm 0.0$ |
| LLSPIN | $0.729 \pm 0.031$ | $4.5 \pm 0.2$ |
| Contextual lasso | $0.498 \pm 0.002$ | $2.9 \pm 0.3$ |

prediction performance. The contextual linear model trails closely in prediction, though it does not offer a similar level of parsimony. While also producing good predictions, the deep neural network and lassonet do not offer the same interpretability. The lasso lags far behind the contextual lasso and other neural network-based methods, suggesting that the contextual features have nonlinear effects.

## B  Grouped explanatory features

Algorithm 2 presents the routine for projecting onto the group $\ell_1$-ball. To summarize the algorithm,

---
**Algorithm 2** Projection onto group $\ell_1$-ball

---

    **input** Dense group coefficients $\boldsymbol{\eta}_1^{(k)}, \dots, \boldsymbol{\eta}_n^{(k)}$ ($k = 1, \dots, g$) and radius $\lambda$

    Compute group-wise norms $\xi_i^{(k)} = \|\boldsymbol{\eta}_i^{(k)}\|_2$ for $i = 1, \dots, n$ and $k = 1, \dots, g$

    Run Algorithm 1 with $\xi_1^{(k)}, \dots, \xi_n^{(k)}$ ($k = 1, \dots, g$) and $\lambda$ to get $\bar{\xi}_1^{(k)}, \dots, \bar{\xi}_n^{(k)}$ ($k = 1, \dots, g$)

    Compute $\boldsymbol{\beta}_1^{(k)}, \dots, \boldsymbol{\beta}_n^{(k)}$ as $\boldsymbol{\beta}_i^{(k)} = \boldsymbol{\eta}_i^{(k)} \bar{\xi}_i^{(k)} / \xi_i^{(k)}$ for $i = 1, \dots, n$ and $k = 1, \dots, g$

    **output** Group-sparse coefficients $\boldsymbol{\beta}_1^{(k)}, \dots, \boldsymbol{\beta}_n^{(k)}$ ($k = 1, \dots, g$)

---

the norm of each group is projected onto the $\ell_1$-ball using Algorithm 1, and then each set of group coefficients is rescaled by the resulting projected norms. These projected norms can be zero after thresholding, yielding sparsity across the groups. For the correctness of Algorithm 2, refer to Theorem 4.1 in van den Berg et al. (2008), which establishes the validity of this type of thresholding.

## C  Side constraints

To simplify notation here, we refer to $\boldsymbol{\eta}(\mathbf{z}_i)$ by the shorthand $\boldsymbol{\eta}_i$. The $\ell_1$-projection with sign constraints is

$$
\min_{\boldsymbol{\beta}_1, \dots, \boldsymbol{\beta}_n} \quad \frac{1}{n} \sum_{i=1}^{n} \|\boldsymbol{\eta}_i - \boldsymbol{\beta}_i\|_2^2
$$

$$
\text{s.t.} \quad \frac{1}{n} \sum_{i=1}^{n} \|\boldsymbol{\beta}_i\|_1 \leq \lambda \tag{6}
$$

$$
\beta_{ij} \geq 0, \; i = 1, \dots, n, \; j \in \mathcal{P}
$$

$$
\beta_{ij} \leq 0, \; i = 1, \dots, n, \; j \in \mathcal{N}.
$$

Here, $\mathcal{P} \subseteq \{1, \dots, p\}$ and $\mathcal{N} \subseteq \{1, \dots, p\}$ index the explanatory features whose coefficients are restricted nonnegative and nonpositive, respectively. Proposition C.1 states that (6) reduces to a simpler problem directly solvable by Algorithm 1.

**Proposition C.1.** *Let* $\boldsymbol{\eta}_1, \ldots, \boldsymbol{\eta}_n \in \mathbb{R}^p$. *Define* $\tilde{\boldsymbol{\eta}}_1, \ldots, \tilde{\boldsymbol{\eta}}_n$ *elementwise as*

$$
\tilde{\eta}_{ij} = \begin{cases} 0 & \text{if } \eta_{ij} < 0 \wedge j \in \mathcal{P} \\ 0 & \text{if } \eta_{ij} > 0 \wedge j \in \mathcal{N} \, , \quad i = 1, \ldots, n, \, j = 1, \ldots, p. \\ \eta_{ij} & \text{otherwise} \end{cases}
$$

*Then optimization problem* (6) *admits the same optimal solution as*

$$
\min_{\boldsymbol{\beta}_1, \ldots, \boldsymbol{\beta}_n} \quad \frac{1}{n} \sum_{i=1}^n \|\tilde{\boldsymbol{\eta}}_i - \boldsymbol{\beta}_i\|_2^2
$$
$$
\text{s. t.} \quad \frac{1}{n} \sum_{i=1}^n \|\boldsymbol{\beta}_i\|_1 \leq \lambda. \tag{7}
$$

The proof of this proposition requires the following lemma.

**Lemma C.2.** *A solution* $\boldsymbol{\beta}_1, \ldots, \boldsymbol{\beta}_n$ *to* (6) *must satisfy the inequality* $\eta_{ij} \beta_{ij} \geq 0$ *for all* $i$ *and* $j$.

*Proof.* We proceed using proof by contradiction along the lines of Lemma 3 in Duchi et al. (2008). Suppose there exists a solution $\boldsymbol{\beta}_1, \ldots, \boldsymbol{\beta}_n$ such that $\eta_{ij} \beta_{ij} < 0$ for some $i$ and $j$. Take $\boldsymbol{\beta}_1^\star, \ldots, \boldsymbol{\beta}_n^\star$ equal to $\boldsymbol{\beta}_1, \ldots, \boldsymbol{\beta}_n$ except at index $(i, j)$, where we set $\beta_{ij}^\star = 0$. Note that $\boldsymbol{\beta}_1^\star, \ldots, \boldsymbol{\beta}_n^\star$ continues to satisfy the sign constraints and $\ell_1$-constraint, and hence remains feasible for (6). We also have that

$$
\sum_{i=1}^n \|\boldsymbol{\eta}_i - \boldsymbol{\beta}_i\|_2^2 - \sum_{i=1}^n \|\boldsymbol{\eta}_i - \boldsymbol{\beta}_i^\star\|_2^2 = \beta_{ij}^2 - 2\eta_{ij}\beta_{ij} > \beta_{ij}^2 > 0.
$$

Thus, $\boldsymbol{\beta}_1^\star, \ldots, \boldsymbol{\beta}_n^\star$ attains a lower objective value than the solution $\boldsymbol{\beta}_1, \ldots, \boldsymbol{\beta}_n$. This contradiction yields the statement of the lemma. $\qquad\square$

The proof of Proposition C.1 now follows.

*Proof.* If $\eta_{ij} < 0$ and $j \in \mathcal{P}$, then by Lemma C.2, a solution to (6) must satisfy $\beta_{ij} = 0$. Likewise, if $\eta_{ij} > 0$ and $j \in \mathcal{N}$, then it must hold that $\beta_{ij} = 0$. Hence, by setting any $\eta_{ij}$ that violate the sign constraints to zero (i.e., $\tilde{\eta}_{ij}$), and noting that a solution to (7) must satisfy $\beta_{ij} = 0$ when $\tilde{\eta}_{ij} = 0$, we arrive at the result of the proposition. $\qquad\square$

# D   Pathwise optimization

In a spirit similar to Friedman et al. (2007), we take the sequence of regularization parameters $\{\lambda^{(t)}\}_{t=1}^T$ as a grid of values that yields a path between the unregularized model (no sparsity) and the fully regularized model (all coefficients zero). Specifically, we set $\lambda^{(1)}$ such that the contextual lasso regularizer does not impart any regularization, i.e., $\lambda^{(1)} = n^{-1} \sum_{i=1}^n \|\boldsymbol{\beta}_{\hat{\mathbf{w}}^{(1)}}(\mathbf{z}_i)\|_1$, where the weights $\hat{\mathbf{w}}^{(1)}$ are a solution to (4) from setting $\lambda = \infty$. We then construct the sequence as a grid of linearly spaced values between $\lambda^{(1)}$ and $\lambda^{(T)} = 0$, the latter forcing all coefficients to zero. Linear spacing of the sequence of $\lambda^{(t)}$ generally yields linearly spaced sparsity levels. The sequence should be decreasing so the optimizer can build on networks that increase in sparsity.

Algorithm 3 summarizes the complete pathwise optimization process, with gradient descent employed as the optimizer. To parse the notation used in the algorithm, $L(\mathbf{w}; \lambda) = n^{-1} \sum_{i=1}^n l(\mathbf{x}_i^\top \boldsymbol{\beta}_{\mathbf{w}}(\mathbf{z}_i), y_i)$ is the loss as a function of the network's weights $\mathbf{w}$ given $\lambda$, and $\nabla_{\mathbf{w}} L(\mathbf{w}; \lambda)$ is its gradient.

# E   Relaxed fit

Denote by $\hat{\boldsymbol{\beta}}_\lambda(\mathbf{z})$ a contextual lasso network fit with regularization parameter $\lambda$. To unwind bias in $\hat{\boldsymbol{\beta}}_\lambda(\mathbf{z})$, we train a polished network $\boldsymbol{\beta}_\lambda^p(\mathbf{z})$ that selects the same explanatory features but does not impose any shrinkage. For this task, we introduce the function $\hat{\mathbf{s}}_\lambda(\mathbf{z}) : \mathbb{R}^m \to \{0, 1\}^p$ that outputs a vector with elements equal to one wherever $\hat{\boldsymbol{\beta}}_\lambda(\mathbf{z})$ is nonzero and elsewhere is zero. We then fit the

---
**Algorithm 3** Pathwise optimization

---
**input** Initial weights $\hat{\mathbf{w}}^{(0)}$, step size $\alpha$, and number of regularization parameters $T$
Initialize $\lambda^{(1)} = \infty$
**for** $t = 1, \ldots, T$ **do**
    Initialize $\mathbf{w}_{(0)} = \hat{\mathbf{w}}^{(t-1)}$
    Initialize $m = 0$
    **while** Not converged **do**
        Update $\mathbf{w}_{(m+1)} = \mathbf{w}_{(m)} - \alpha \cdot \nabla_{\mathbf{w}} L(\mathbf{w}_{(m)}; \lambda^{(t)})$
        Update $m = m + 1$
    **end while**
    Set $\hat{\mathbf{w}}^{(t)} = \mathbf{w}_{(m)}$
    **if** $t = 1$ **then**
        Set $\lambda^{(1)} = n^{-1} \sum_{i=1}^{n} \|\boldsymbol{\beta}_{\hat{\mathbf{w}}^{(1)}}(\mathbf{z}_i)\|_1$ and $\lambda^{(T)} = 0$
        Equispace $\lambda^{(2)}, \ldots, \lambda^{(T-1)}$ between $\lambda^{(1)}$ and $\lambda^{(T)}$
    **end if**
**end for**
**output** Fitted weights $\hat{\mathbf{w}}^{(1)}, \ldots, \hat{\mathbf{w}}^{(T)}$

---

polished network as $\boldsymbol{\beta}_\lambda^p(\mathbf{z}) = \boldsymbol{\eta}(\mathbf{z}) \circ \hat{\mathbf{s}}_\lambda(\mathbf{z})$, where $\circ$ means element-wise multiplication and $\boldsymbol{\eta}(\mathbf{z})$ is the same architecture as used for the original contextual lasso network before the projection layer. The effect of including $\hat{\mathbf{s}}_\lambda(\mathbf{z})$, which is fixed when training $\boldsymbol{\beta}_\lambda^p(\mathbf{z})$, is twofold. First, it guarantees the coefficients from the polished network are nonzero in the same positions as the original network, i.e., the same features are selected. Second, it ensures explanatory features only contribute to gradients for observations in which they are active, i.e., $x_{ij}$ does not contribute if the $j$th component of $\hat{\mathbf{s}}_\lambda(\mathbf{z}_i)$ is zero. Because the polished network does not project onto an $\ell_1$-ball, its coefficients are not shrunk.

To arrive at the relaxed contextual lasso fit, we combine $\hat{\boldsymbol{\beta}}_\lambda(\mathbf{z})$ and the fitted polished network $\hat{\boldsymbol{\beta}}_\lambda^p(\mathbf{z})$:

$$\hat{\boldsymbol{\beta}}_{\lambda,\gamma}(\mathbf{z}) := (1-\gamma)\hat{\boldsymbol{\beta}}_\lambda(\mathbf{z}) + \gamma\hat{\boldsymbol{\beta}}_\lambda^p(\mathbf{z}), \quad 0 \leq \gamma \leq 1. \tag{8}$$

When $\gamma = 0$, we recover the original biased coefficients, and when $\gamma = 1$, we attain the unbiased polished coefficients. Between these extremes lies a continuum of relaxed coefficients with varying degrees of bias. Since the original and polished networks need only be computed once, we may consider any relaxation on this continuum at virtually no computational expense over and above that of the two networks. In practice, we choose among the possibilities by tuning $\gamma$ on a validation set.

To illustrate the benefits of relaxation, we present Figure 6, which compares the relaxed and nonrelaxed variants of the contextual lasso under the synthetic experimental design of Section 4. As far as prediction accuracy is concerned, the benefits of relaxation are marginal. However, for selection and interpretation, the story is quite different. The relaxation yields models that are both sparser and contain more true positives and/or fewer false positives. These gains are most pronounced in larger samples. In smaller samples, relaxation is less beneficial because the bias from shrinkage helps stabilize the models. Yet, the relaxed variant of the contextual lasso typically does no worse than its nonrelaxed counterpart because it adapts to the best level of bias (shrinkage) by tuning $\gamma$.

## F  Run times

The complexity analysis in Section 2.7 suggests that the training time for the contextual lasso should be linear in the sample size $n$ and number of explanatory features $p$. To verify this result, we record the time taken to fit the contextual lasso over a sequence of 50 values of $\lambda$. The number of hidden layers is three and the number of neurons per layer is 100. Figure 7 reports the results. The run times are indeed linear as a function of $n$ and $p$ and, overall, quite reasonable for real world applications.

## G  Localized lasso

We compare the contextual lasso with the localized lasso of Yamada et al. (2017). The graph information required by the localized lasso is estimated from the contextual features using the nearest neighbors approach of Yamada et al. (2017). We focus on smaller sample sizes than in the main

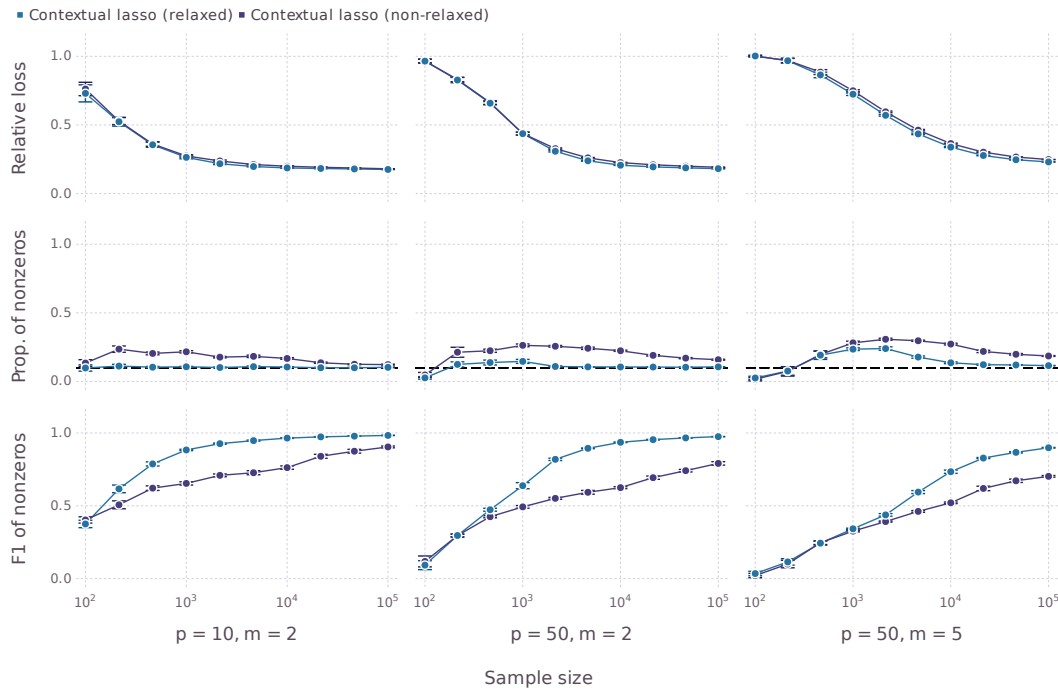

Figure 6: Relaxation ablation on synthetic regression data. Metrics are aggregated over 10 synthetic datasets. Solid points are averages and error bars are standard errors. Dashed horizontal lines in the middle row indicate the true sparsity level.

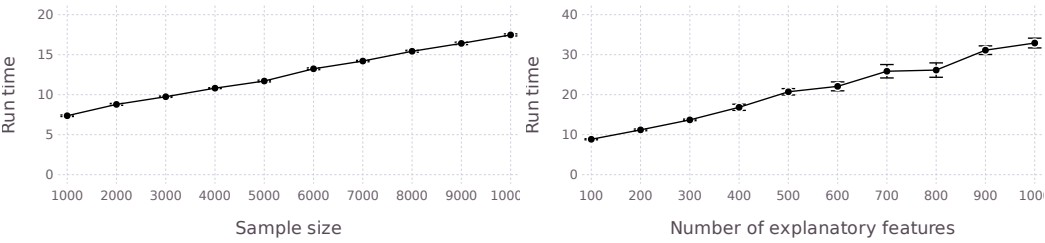

Figure 7: Run time in seconds to fit the contextual lasso over a sequence of 50 values of $\lambda$, measured over 10 synthetic datasets. The number of explanatory features $p = 50$ in the left plot and the sample size $n = 1,000$ in the right plot. The number of contextual features $m = 5$. Solid points are averages and error bars are one standard errors.

experiments since each observation requires a new coefficient vector that each constitute additional optimization variables. Figure 8 reports the results for regression (the localized lasso does not support classification). The localized lasso's prediction loss improves with growing $n$, but not nearly as well as the contextual lasso. The graph information may be insufficient to encode the underlying nonlinearity. Furthermore, the localized lasso's regularizer never induces fully sparse coefficient vectors (all zeros), which may be limiting if there are no relevant explanatory features for certain $\mathbf{z}$.

# H   Implementation details

The contextual lasso is fit using our `Julia` package `ContextualLasso`. The network is configured with three hidden layers. The number of neurons, which are spread equally across these hidden layers, is set so that the dimensionality of the weights $\mathbf{w}$ is approximately $32 \times p \times m$. This setting ensures the network size scales roughly linearly with the number of features. The contextual linear model uses the same architecture, excluding the projection layer. The deep neural network is set up similarly. These methods all use rectified linear activation functions in the hidden layers and are optimized

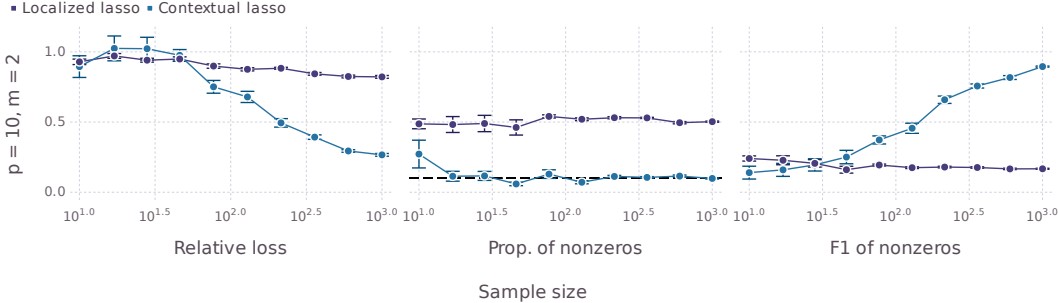

Figure 8: Comparisons with the localized lasso on synthetic regression data. Metrics are aggregated over 10 synthetic datasets. Solid points are averages and error bars are standard errors. The dashed horizontal line in the middle indicates the true sparsity level.

using Adam (Kingma and Ba, 2015) with a learning rate of 0.001. Convergence is monitored on a validation set with the optimizer terminated after 30 iterations without improvement.

The lasso is fit using the `Julia` package `GLMNet` (Friedman et al., 2010). The group lasso is fit using the `R` package `grpreg` (Breheny and Huang, 2015). Since contextual features are always relevant, the regularizer for the lasso is applied only to explanatory features and interactions, not the contextual features.[9] The (group) lasso and contextual lasso all allow for relaxed fits, as discussed in Section 2.6. The regularization parameter $\lambda$ is swept over a grid of 50 values computed automatically from the data using `ContextualLasso`, `GLMNet`, or `grpreg`. For each value of $\lambda$, the relaxation parameter $\gamma$ is swept over the grid $\{0, 0.1, ..., 1\}$.

The lassonet is fit using the `Python` package `lassonet` (Lemhadri et al., 2021b), which also performs its own relaxation. LLSPIN is fit using the authors' `Python` implementation (see Yang et al., 2022). Their implementation relies on the hyperparameter optimization framework `Optuna` (Akiba et al., 2019) to determine the regularization parameter and learning rate. We use the default grid for tuning the learning rate, but increase the maximum regularization parameter to 10, which is roughly the smallest value required to achieve a fully sparse solution in our experiments. LLSPIN does not shrink and so does not admit a relaxation. Lassonet and LLSPIN use the same convergence criterion, number of regularization parameters, and network configuration as the other deep learning methods.

For all methods, the input features are standardized prior to training. Standardization of the explanatory features is necessary for the lasso estimators as it places all coefficients on the same scale, ensuring equitable regularization. `ContextualLasso` automates standardization and expresses all final coefficients on their original scale.

All experiments are run on a Linux platform with NVIDIA RTX 4090 GPUs.

## I   Synthetic data generation

The explanatory features $\mathbf{x}_1, \ldots, \mathbf{x}_n$ are generated iid as $p$-dimensional $N(\mathbf{0}, \mathbf{\Sigma})$ random variables, where the covariance matrix $\mathbf{\Sigma}$ has elements $\Sigma_{ij} = 0.5^{|i-j|}$. The contextual features $\mathbf{z}_1, \ldots, \mathbf{z}_n$ are generated iid as $m$-dimensional random variables uniform on $[-1, 1]^m$, independent of the $\mathbf{x}_i$. With the features drawn, we simulate a regression response as

$$y_i \sim N(\mu_i, 1), \quad \mu_i = \kappa \cdot \mathbf{x}_i^\top \boldsymbol{\beta}(\mathbf{z}_i),$$

or a classification response as

$$y_i \sim \text{Bernoulli}(p_i), \quad p_i = \frac{1}{1 + \exp\left(-\kappa \cdot \mathbf{x}_i^\top \boldsymbol{\beta}(\mathbf{z}_i)\right)},$$

for $i = 1, \ldots, n$. Here, $\kappa > 0$ controls the signal strength vis-à-vis the variance of $\kappa \cdot \mathbf{x}_i^\top \boldsymbol{\beta}(\mathbf{z}_i)$. We first estimate the variance of $\mathbf{x}_i^\top \boldsymbol{\beta}(\mathbf{z}_i)$ on the training set and then set $\kappa$ so the variance of the signal

---

[9] `grpreg` does not provide support for this functionality.

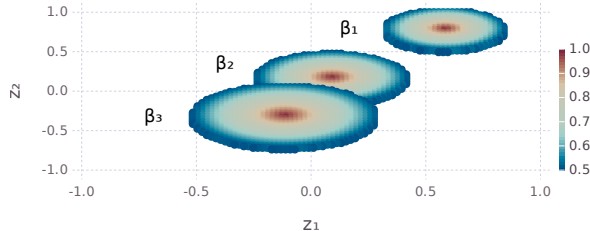

Figure 9: Illustration of coefficient function (9) for $p = 3$ explanatory features and $m = 2$ contextual features. The centers $\mathbf{c}_j$ correspond to the dark red in the middle of each sphere.

is five. The coefficient function $\boldsymbol{\beta}(\mathbf{z}) := \big(\beta_1(\mathbf{z}), \ldots, \beta_p(\mathbf{z})\big)^\top$ is constructed such that $\beta_j(\mathbf{z}_i)$ maps to a nonzero value whenever $\mathbf{z}_i$ lies within a hypersphere of radius $r_j$ centered at $\mathbf{c}_j$:

$$\beta_j(\mathbf{z}_i) = \begin{cases} 1 - \frac{1}{2r_j} \|\mathbf{z}_i - \mathbf{c}_j\|_2 & \text{if } \|\mathbf{z}_i - \mathbf{c}_j\|_2 \leq r_j \\ 0 & \text{otherwise} \end{cases}. \tag{9}$$

This function attains the maximal value one when $\mathbf{z}_i = \mathbf{c}_j$ and the minimal value zero when $\|\mathbf{z}_i - \mathbf{c}_j\|_2 > r_j$. The centers $\mathbf{c}_1, \ldots, \mathbf{c}_p$ are generated with uniform probability on $[-1, 1]^p$, and the radii $r_1, \ldots, r_p$ are chosen to achieve sparsity levels that vary between 0.05 and 0.15 (average 0.10). Figure 9 provides a visual illustration. This function is inspired by the house pricing example in Figure 1, where the coefficients are nonzero in some central region of the contextual feature space.

## J    Classification results

### J.1    Synthetic data

Figure 10 reports the experimental results for classification on synthetic data, analogous to those for regression in Section 4. The findings are broadly in line with the regression ones. The contextual lasso performs best overall and is the only method ever able to recover the true nonzeros accurately.

### J.2    News popularity data

We turn to a real dataset of articles posted to the news platform Mashable (Fernandes et al., 2015). The task is to predict if an article will be popular, defined in Fernandes et al. (2015) as more than 1400 shares. In addition to the zero-one response feature for popularity, the dataset has predictive features that quantify the articles (e.g., number of total words, positive words, and images). The data channel feature, which identifies the category of the article (lifestyle, entertainment, business, social media, technology, world, or viral), is taken as the contextual feature. It is expressed as a sequence of indicator variables yielding $m = 6$ contextual features. There remain $p = 51$ explanatory features.

Table 4 reports the results over 10 random splits of the dataset's $n = 39,643$ observations into training, validation, and testing sets in 0.6-0.2-0.2 proportions. In contrast to the previous datasets, all

Table 4: Comparisons on the news popularity data. Metrics are aggregated over 10 random splits of the data. Averages and standard errors are reported.

|  | Relative loss | Avg. sparsity |
| --- | --- | --- |
| Deep neural network | $0.903 \pm 0.003$ | $51.0 \pm 0.0$ |
| Contextual linear model | $0.906 \pm 0.003$ | $51.0 \pm 0.0$ |
| Lasso | $0.914 \pm 0.002$ | $22.4 \pm 0.7$ |
| Lassonet | $0.894 \pm 0.002$ | $50.7 \pm 0.3$ |
| LLSPIN | $0.923 \pm 0.009$ | $51.0 \pm 0.0$ |
| Contextual lasso | $0.906 \pm 0.002$ | $12.6 \pm 0.9$ |

methods predict similarly well here. The lassonet performs marginally best overall, while LLSPIN performs marginally worst. Though predicting neither best nor worst, the contextual lasso retains a

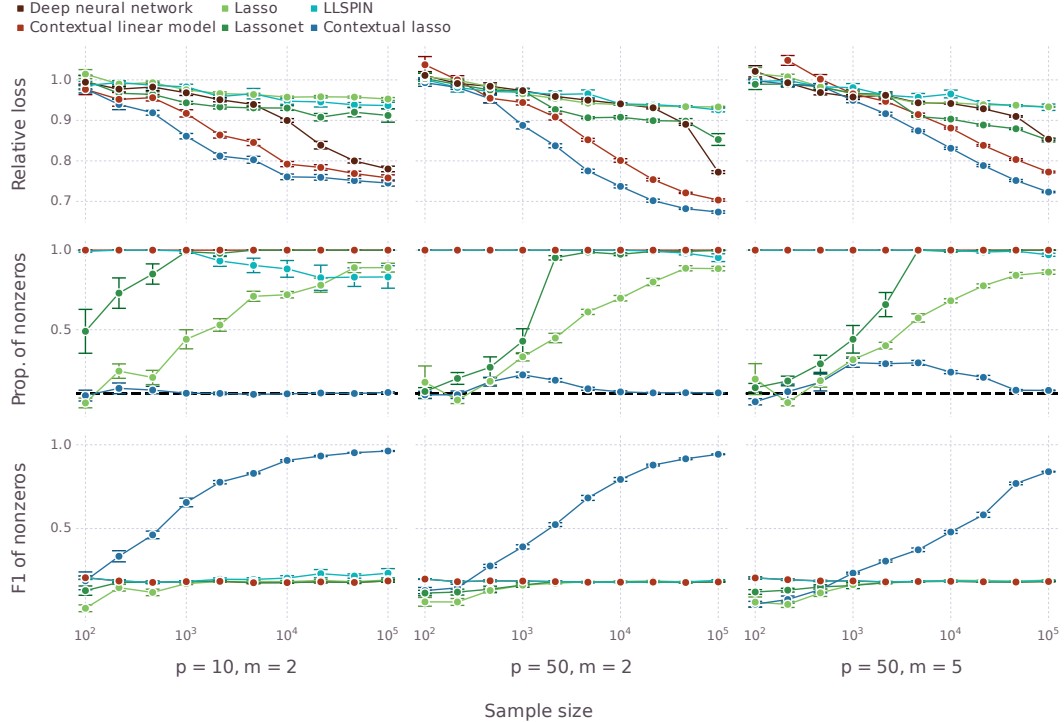

Figure 10: Comparisons on synthetic classification data. Metrics are aggregated over 10 synthetic datasets. Solid points are averages and error bars are standard errors. Dashed horizontal lines in the middle row indicate the true sparsity level.

significant lead in terms of sparsity, being twice as sparse as the next sparsest method (lasso). Sparsity is crucial for this task as it allows the author or editor to focus on a small number of changes necessary to improve the article's likelihood of success. Other methods such as the uninterpretable deep neural network, or fully dense contextual linear model, are not nearly as useful for the same purpose.

# K   High-dimensional and fixed coefficient results

Besides interpretability, a major appeal of the lasso is its good performance in high-dimensional regimes, where the number of features is comparable to the sample size. It is intriguing to consider whether the contextual lasso remains useful in this setting. To this end, we extend the experiments of Section 4 to $p = 1,000$ explanatory features. Typically, when there are so many features, only a small number are relevant for predicting the response, so we adjust the synthetic data generation process so that only 10 explanatory features are relevant for some values of the contextual features $\mathbf{z}$. The remaining 990 explanatory features remain irrelevant for all $\mathbf{z}$. Figure 11 reports the results. The contextual lasso performs highly competitively, even when $n < 1,000$ and there are more explanatory features than observations. As with the lower-dimensional experiments, the contextual lasso can still filter out the irrelevant explanatory features and achieves near-perfect support recovery for large $n$.

A second regime that might arise in practice is where contextual features are present but have no effect on the explanatory features. That is, the explanatory features have a fixed sparsity pattern and fixed coefficients for all $\mathbf{z}$. Figure 12 reports the results in this setting. Here, the contextual lasso is outperformed by the lasso (which is in its home territory) and the lassonet, both of which assume the sparsity pattern is fixed. Nonetheless, the contextual lasso is able to recover the true nonzeros by learning a constant function for $\boldsymbol{\beta}(\mathbf{z})$ via the bias term in each neuron. Provided the sample size is sufficiently large, it is reasonable to expect the contextual lasso to remain competitive with the lasso.

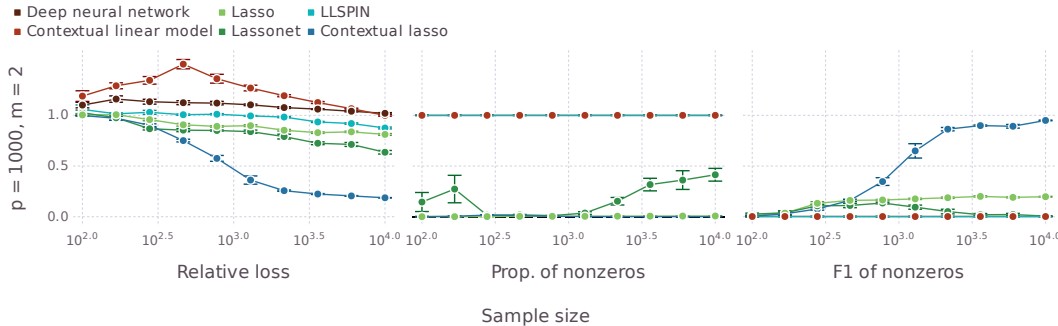

Figure 11: Comparisons on high-dimensional synthetic regression data. Metrics are aggregated over 10 synthetic datasets. Solid points are averages and error bars are standard errors. The dashed horizontal line in the middle indicates the true sparsity level.

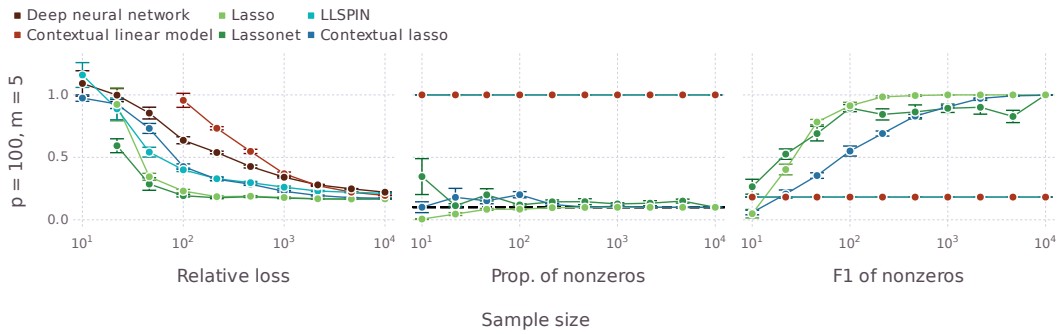

Figure 12: Comparisons on fixed coefficient synthetic regression data. Metrics are aggregated over 10 synthetic datasets. Solid points are averages and error bars are standard errors. The dashed horizontal line in the middle indicates the true sparsity level.

## L  Stability analysis

Since the contextual lasso involves a highly-parameterized neural network, it is insightful to consider its selection stability when trained using different random initializations of the network weights. To this end, we report Figure 13, which shows selection instability as measured by the Hamming distance (scaled by $p$) between two independently initialized networks. Unsurprisingly, the statistically difficult

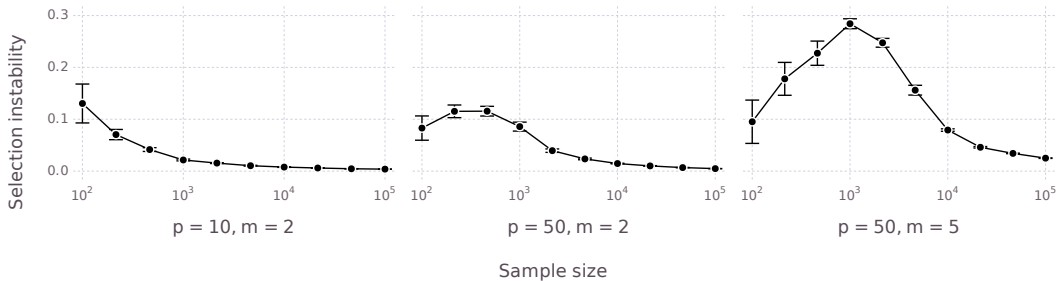

Figure 13: Selection instability of the contextual lasso over 10 synthetic datasets. Solid points represent averages and error bars denote standard errors.

regimes, where the sample size is small or the number of explanatory/contextual features is large, correspond to higher instability. In any case, the contextual lasso grows increasingly stable with the sample size. Even when $p = 50$ and $m = 5$, the contextual lasso is almost entirely stable by the time $n = 10,000$, which is typical of our real data examples. The concave shape of the stability curve

when $p = 50$ is consistent with the proportion of nonzeros reported in Figure 3, where the contextual lasso initially produces highly sparse models that are more restricted and hence more stable.

## M    Dataset availability

The datasets used throughout this paper are publicly available at the following URLs.

- House pricing data in Section 1:
  https://www.kaggle.com/datasets/ruiqurm/lianjia.
- Energy consumption data in Section 4:
  https://archive.ics.uci.edu/ml/datasets/Appliances+energy+prediction.
- Parkinson's telemonitoring data in Section 4:
  https://archive.ics.uci.edu/ml/datasets/parkinsons+telemonitoring.
- News popularity data in Appendix J:
  https://archive.ics.uci.edu/ml/datasets/online+news+popularity.

## N    Limitations

The contextual lasso has strong prediction, interpretation, and selection properties. Naturally, it also has several weaknesses. As with the ordinary lasso, the contextual lasso may select a single feature from a group of highly correlated features. This shortcoming could be remedied by introducing an $\ell_2^2$-regularizer in a spirit similar to the elastic net (Zou and Hastie, 2005). Another potential drawback is that the contextual lasso does not guarantee the complete exclusion of an explanatory feature. That is, it cannot ensure the coefficient for a feature is nonzero for every possible $\mathbf{z}$, which might limit interpretability in certain settings. Full exclusion could be achieved by adding an $\ell_1$-regularizer on the weights of the first layer, similar to the lassonet. Finally, as is typical of neural networks, the objective function of the contextual lasso is nonconvex, which can complicate analysis of its properties.

