# OpenReview forum: "The Contextual Lasso: Sparse Linear Models via Deep Neural Networks"
_NeurIPS.cc/2023/Conference — NeurIPS 2023 poster_

### Official Review · Reviewer_kkJW · 2023-07-01

**Soundness:** 3 good
**Presentation:** 3 good
**Contribution:** 2 fair
**Rating:** 4
**Confidence:** 3

**Summary:**

This paper proposes the contextual lasso, considering the contextual sparsity which allows feature relevance to depend on the prediction context.To solve the optimization problem of contextual lasso, deep neural networks is utilized to learn interpretable sparse linear models. Besides, an extensive experimental analysis of the new estimator illustrates its good prediction, interpretation, and selection properties in various settings.

**Strengths:**

1. The main novelty of this paper is that it simultaneously considers the explanatory feature and contextual feature in the classical lasso framework. In this case, the classical lasso can be viewed as a special case of contextual lasso.
2. The performance compared to lasso shows that combining the contextual feature and explanatory feature as proposed in the paper leads to performance benefits.
3. The paper is well-written and technically sound.

**Weaknesses:**

1. An essential problem of this paper is that how to accurately define or distinct the explanatory feature and contextual feature still remains unclear. The authors should offer more definition or evidence to explain the contextual feature.
2. The comparison methods are insufficient. Especially, since contextual explanation networks is a cousin of the contextual lasso, it is necessary to show the superiority of contextual lasso against contextual explanation networks (Al-Shedivat et al.,2020).
3. Though some experiments can validate the effectiveness of contextual lasso over lasso, it also lacks a time complexity analysis.
4. Theoretical analysis (e.g., generalization, optimization) seems weak  for the proposed approach.


**Questions:**

1. The authors should provide detailed explanation or definition about the contextual feature. Moreover, how to achieve the additional contextual feature from the real-world data should also be included.
2. The authors should add more comparison methods including the contextual explanation networks.
3. The authors should offer the time complexity analysis of contextual lasso.
4. At least, some discussions are necessary for the theoretical properties of contextual lasso, e.g., generalization ability compared with the previous Lasso-type methods.

**Limitations:**

One of the limitations of contextual lasso referred by authors is that it can result in bias of the linear model coefficients towards zero. This drawback is also due to the L1-norm and indeed belongs to all lasso estimators. The authors propose a possible relaxing approach.

---

> ### Author Rebuttal · Authors · 2023-08-10
>
> 1. **Comment:** The authors should provide detailed explanation or definition about the contextual feature. Moreover, how to achieve the additional contextual feature from the real-world data should also be included.
> **Response:** We will add the following paragraph to the paper's conclusion:
> > In practice, one must decide on the split into explanatory and contextual features. The problem of deciding on this split is the same as that faced with varying-coefficient models. Though the literature on varying-coefficient models is extensive, there are no definitive rules for partitioning the features in general. In the housing and energy examples, the contextual features are temporal or spatial effects, which are distinct from the remaining features. In the telemonitoring example, the patient attributes (age and sex) differ fundamentally from the vocal characteristics, so the delineation is clear there. Ultimately, the partition for any given application should be guided by domain expertise with consideration to the end goal. If the user needs to understand the exact effect of a feature, that feature should be an explanatory feature. If a feature's effect is of secondary interest, or it is suspected that the feature influences the structural form of the model, that feature should be a contextual feature. If the user determines there are no contextual features, the ordinary lasso is a more appropriate tool.
> 2. **Comment:** The authors should add more comparison methods including the contextual explanation networks.
> **Response:** The contextual linear model evaluated in our paper is the contextual explanation network with deterministic encoding and linear explanation in Al-Shedivat, Dubey, and Xing (2020). Please see "varying-coefficient models" (Section 5.1) and "deterministic encoding" (Section 3.1.1) of that paper. They also consider a "constrained deterministic encoding" (Section 3.1.2), which involves soft-attention over a finite-set of linear models (described in the literature review of our paper). We found this variant did not perform as well during the development of our approach. We will revise the related works section of our paper to clarfiy the connection. As an additional benchmark, we now also evaluate the localized lasso (Yamada et al. 2017). The localized lasso requires graph information, which we extract from the contextual features. It's prediction loss improves with growing $n$, but at a slower rate than the contextual lasso. Please see Figure 3 of the PDF.
> 2. **Comment:** The authors should offer the time complexity analysis of contextual lasso.
> **Response:** A forward/backward pass through the vanilla feedforward component of the network takes $O(md+hd^2+pd)$ time, where $h$ is the number of hidden layers, $d$ is the number of nodes per layer, $m$ is the number of contextual features, and $p$ is the number of explanatory features. A forward/backward pass through the projection algorithm takes $O(p)$ time (Duchi et al. 2008). Over a sample of size $n$, these become $O(nd(m+hd+p))$ and $O(np)$, respectively. The time complexity for the full network over $n$ observations is thus $O(nd(m+hd+p))$. We will add these details to the paper. We also added runtime experiments that empirically demonstrate linear complexity as a function of $n$ and $p$. Please see Figure 5 of the PDF. As a point of comparison, in the left plot at $n=10,000$, Lassonet takes $\sim50$ seconds and LLSPINN takes $\sim470$ seconds. In the right plot, at $p=1,000$, Lassonet takes $\sim35$ seconds and LLSPINN takes $\sim95$ seconds. The run time for other methods is negligible ($\sim 1$ second).
> 3. **Comment:** At least, some discussions are necessary for the theoretical properties of contextual lasso, e.g., generalization ability compared with the previous Lasso-type methods.
> **Response:** We will add the following paragraph to the paper:
> > The statistical properties of the lasso in terms of estimation, prediction, and selection are now well-established in theory  (Bunea, Tsybakov, and Wegkamp 2007; Raskutti, Wainwright, and Yu 2011; Shen et al. 2013; Zhang, Wainwright, and Jordan 2014). The synthetic experiments in our paper suggest that the contextual lasso satisfies similar properties, though theoretically establishing these results is challenging. Convergence results for vanilla feedforward neural networks (e.g., Schmidt-Hieber 2020) do not apply in our setting due to the projection layer. Moreover, to our knowledge, no statistical guarantees exist for networks configured with convex optimization layers in general. It remains an important avenue of future research to establish a solid theoretical foundation for the contextual lasso.

---

> > ### Comment · Reviewer_kkJW · 2023-08-19
> > **Thanks.**
> >
> > I would like to thank the authors for answering my questions. I am keeping my original score.

---

### Official Review · Reviewer_DGcD · 2023-07-04

**Soundness:** 3 good
**Presentation:** 3 good
**Contribution:** 3 good
**Rating:** 6
**Confidence:** 2

**Summary:**

This paper proposed the contextual lasso model to combine the interpretability of linear model and the express power of deep neural network. The input feature are split into explanatory features and contextual features. The explanatory features are used as input features for a generalized linear model, and the contextual features are feed into a neural network with a projection layer on top to compute the parameters of the linear model. The projection layer are formed as a constrained optimization to force the linear model parameters lie in a $L_1$ ball. Grouped features and other extensions are also discussed. Experiment results show the proposed model outperform traditional DNN model and contextual linear model without LASSO sparsity.

**Strengths:**

The paper is well written and easy to follow. Using contextual features to compute parameters for an interpretable sparse linear model seems to be an interesting and novel idea. Many extensions are discussed. The empirical results also support that the proposed model can select sparse model while maintain better prediction performance.

**Weaknesses:**

A straightforward idea for a sparse DNN model is to include all explanatory features and contextual features as input to a DNN, train the model with some regularizations, after some thresholding, check the weights in the first layer(or multiplication of indicator matrix for all layers) to decide whether a variable is select or not. With appropriate regularization, I think this can also select relevant variables, and the variable importance can be measured by the gradient to the variable. Conceptually, why does the contextual model better? For real data, how should we decide which features are explanatory features or contextual features?

And empirically, It was a bit surprise for me that the baseline Lassonet always keep a relatively dense model, is it due to some poor hyper-parameter choice? please see the Questions section for some clarification.

**Questions:**

1. The projection layer works on the average coefficient vector, how does it work with mini-batch training for DNN, does it just look at the coefficient in that mini-batch?

2. In the experiments, for DNN type model, does contextual features included in the input? In line 280, the DNN and Bassinet are applied to the original features, it might be beneficial to include results of DNN or lassonet with all spline features and contextual features as input

3. Looking at the results of lassonet, it almost select all variables for Parkinson and Energy data sets and select all variables when n gets large for simulation data. How is the regularization parameter determined for lassonet. If we chose a large regularization parameter manually to force the model to have similar sparsity level as the contextual lasso, would the prediction performance be much worse?

---

> ### Author Rebuttal · Authors · 2023-08-10
>
> 1. **Comment:** A straightforward idea for a sparse DNN model is to include all explanatory features and contextual features as input to a DNN, train the model with some regularizations, after some thresholding, check the weights in the first layer(or multiplication of indicator matrix for all layers) to decide whether a variable is select or not. With appropriate regularization, I think this can also select relevant variables, and the variable importance can be measured by the gradient to the variable. Conceptually, why does the contextual model better?
> **Response:** Your idea is related to the lassonet, which inputs all features (explanatory and contextual) and thresholds weights on the first layer (using a lasso penalty) to select variables. This approach underperforms relative to the contexutal lasso because the sparsity pattern in our real/synthetic experiments varies, whereas the weights (which are the basis for thresholding) are fixed. Another drawback is limited interpretability because all features pass through the network.
> 2. **Comment:** For real data, how should we decide which features are explanatory features or contextual features?
> **Response:** Please see our response to Comment 1 of Reviewer kkJW---the same question is raised there.
> 3. **Comment:** The projection layer works on the average coefficient vector, how does it work with mini-batch training for DNN, does it just look at the coefficient in that mini-batch?
> **Response:** We use full batch training in our experiments because all datasets readily fit in GPU memory, even those with $n=10^5$ observations. In any case, the projection layer indeed applies with mini batch training by running the projection algorithm on each mini batch. The drawback is that the average coefficient vector is noisier to estimate than with the full batch. We will add the details to the paper.
> 4. **Comment:** In the experiments, for DNN type model, does contextual features included in the input? In line 280, the DNN and Bassinet are applied to the original features, it might be beneficial to include results of DNN or lassonet with all spline features and contextual features as input
> **Response:** Yes, the DNN receives all features (contextual and explanatory) as inputs. We will clarify this point in the manuscript. We reran the results for the Parkinson's data with the DNN and lassonet receiving all explanatory splines and contextual features.
>
>    |                            | Relative loss      | Avg. sparsity      |
>    |----------------------------|--------------------|--------------------|
>    | Deep neural network (w/ spline) | $0.638\pm0.077$ | $16.0\pm0.0$   |
>    | Lassonet (w/ spline)       | $0.335\pm0.007$   | $15.5\pm0.2$   |
>
>    The spline versions witness a decline in performance relative to the existing non-spline versions (compare with Table 2 in the paper). The neural networks can approximate splines themselves, so the additional dimensionality from the splines likely introduces further noise. We will state in the text that using splines with these methods does not improve their performance.
>
> 5. **Comment:** Looking at the results of lassonet, it almost select all variables for Parkinson and Energy data sets and select all variables when n gets large for simulation data. How is the regularization parameter determined for lassonet. If we chose a large regularization parameter manually to force the model to have similar sparsity level as the contextual lasso, would the prediction performance be much worse?
> **Response:** The fact that lassonet often uses all the available features was initially a surprise to us too. However, this behavior seems consistent with the results in Lemhadri et al. (2021), where the best MSE or classification accuracy is typically attained at a dense model (see Figures 1-5 and 7 in that paper). As for the regularization parameter, it is tuned on a validation set in the same manner as the lasso and contextual lasso (see lines 484-491 in the appendix). Manually setting $\lambda$ to achieve a sparsity level similar to the contextual lasso substantially degrades performance on the energy data: the prediction loss is 0.804, roughly twice that from the tuned $\lambda$.

---

> > ### Comment · Reviewer_DGcD · 2023-08-14
> >
> > I appreciate authors for detailed response.
> >
> > Response 2
> >
> > For the explanatory features or contextual features, reading the rebuttal, my understanding is that when the data exhibit such a explanatory\contextual structure, the proposed method would work well. Within this realm, with the additional experiments, I think the paper makes reasonable contribution.
> >
> > Response 5
> >
> > In my experience, I agree that best MSE or classification accuracy often attained at a dense model, so using validation set to select $\lambda$ can result in a dense model. So including results with similar sparsity level will also help demonstrate the advantage of proposed method.
> >
> > I am keeping my score for now and await for reviewer discussion.

---

### Official Review · Reviewer_mL1j · 2023-07-06

**Soundness:** 2 fair
**Presentation:** 3 good
**Contribution:** 2 fair
**Rating:** 5
**Confidence:** 4

**Summary:**

This paper considers the predictive problem setting in which the input features are dichotomized into two groups: explanatory features and contextual features. For such a setting, the authors propose the contextual lasso, a statistical estimator that leverages the expressiveness of deep neural networks to learn interpretable sparse linear models with sparsity patterns that can vary based on the contextual features. More precisely, the authors proposed using a neural network to model the nonlinear effects of the contextual features on the explanatory features. In addition, to enforce sparsity among the coefficients for the explanatory features, the network is trained with a regularizer in the form of a projection layer that maps the output (dense) to sparse coefficients by performing a projection onto the $\ell_1$-ball. Experiments on real and synthetic data suggest that the contextual lasso leads to models which remain interpretable and can be sparser than the regular lasso, without sacrificing the predictive accuracy of standard deep neural networks.

**Strengths:**

This work appears to be the first to propose a sparsity-inducing projection layer that maps a dense output of a neural network onto a space of $\ell_1$-constrained linear models by solving a constrained quadratic program.

The benefit from the proposed estimator is that it can utilize the expressive power of deep neural networks and yet learn interpretable sparse linear models by encouraging contextual sparsity through an additional projection layer; all while allowing feature relevance to depend on the prediction context as opposed to determining a fixed set of relevant features. In addition to the classical contextual sparse learning setting, a variant of the sparsity-encouraging projection layer is also devised for the group lasso setting.
The proposed estimator and the formulation of the quadratic program for determining the sparse coefficients are technically sound and well-supported. Moreover, the particular sparse learning setting that is being considered is well-supported by examples where the input features are inherently dichotomized into explanatory features and contextual features, and the need for contextual sparsity in such a setting is well-justified.

The effectiveness of the contextual lasso has been evaluated on both synthetic and real data (in the standard setting and a setting with grouped explanatory features). The findings seem to suggest that, in contrast to other sparse linear models and a conventional DNN, which use either all or almost all available explanatory features, contextual lasso leads to linear models that are much simpler and involve less explanatory features on average.

The paper is rather well written and organized. The notation is very clear, easy to follow, unambiguous and consistent throughout the paper.

**Weaknesses:**

The problem considered in this work is a rather general one, as many applications involve a certain context that can be leveraged to select candidate features and determine their effects. Nevertheless, this generality of the problem is not matched by the extensiveness of the conducted experiments. It seems that the contribution’s potential significance is limited by (1) the narrow settings of the experimental design (particularly in the synthetic data generation process) and (2) the consideration of only two real-world applications given the pervasiveness of sparse learning problems surrounded by a certain prediction context. I would encourage the authors to consider additional datasets characterized by both explanatory and contextual features.

Although this work is aimed at capturing the prediction context by leveraging the expressive power of neural networks to map contextual features to a sparse coefficient vector, there are other works that take different approaches to the same end. For instance, [1] proposed adaptive optimizers for structured sparsity for deep learning, but also discusses unstructured sparsity ($\ell_1$) which is directly relevant to this work. There are also other works, such as [2], that take a non-neural based approach to capturing prediction context but I believe are still relevant. Namely, [2] addresses the house pricing problem (provided as an example in Section 1 of this work) using a generalization of the group lasso to a graph setting, where the graph represents the context defined by the distances between the house locations. Finally, there are also approaches that are not directly applicable to the setting considered in this paper, however, in a more general sense, they are relevant to inducing sparsity in neural networks and should be considered as a part of the related work (e.g., [3]).

[1] Deleu, T., & Bengio, Y. (2021). Structured sparsity inducing adaptive optimizers for deep learning. arXiv preprint arXiv:2102.03869.
[2] Hallac, D., Leskovec, J., & Boyd, S. (2015, August). Network lasso: Clustering and optimization in large graphs. In Proceedings of the 21th ACM SIGKDD international conference on knowledge discovery and data mining (pp. 387-396).
[3] Wang, P., He, X., Li, G., Zhao, T., & Cheng, J. (2020, April). Sparsity-inducing binarized neural networks. In Proceedings of the AAAI conference on artificial intelligence (Vol. 34, No. 07, pp. 12192-12199).

As previously suggested, other sparse linear estimator variants have been proposed to utilize contextual information for the same or similar setting as the one considered in this work. Such models include Network Lasso (mentioned in the previous comment), Graphical Lasso and its associated extensions, among others. In that regard, I am wondering why the authors did not consider including such estimators as additional baselines. Moreover, since this work proposes a separate projection layer to induce sparsity in the output of a neural network, I also believe that a comparison to a conventional neural network with a dropout layer applied on its outputs ($\eta$ values) would be appropriate.

**Questions:**

My questions/suggestions are provided together with the weaknesses of the paper (where applicable and necessary).

**Limitations:**

A separate “Limitations” section has not been included in this paper. No potential negative societal impact has been declared in the main manuscript. For my improvement suggestions, please refer to the “Weaknesses” section of this review.

---

> ### Author Rebuttal · Authors · 2023-08-10
>
> 1. **Comment:** The problem considered in this work is a rather general one, as many applications involve a certain context that can be leveraged to select candidate features and determine their effects. Nevertheless, this generality of the problem is not matched by the extensiveness of the conducted experiments. It seems that the contribution’s potential significance is limited by (1) the narrow settings of the experimental design (particularly in the synthetic data generation process) and (2) the consideration of only two real-world applications given the pervasiveness of sparse learning problems surrounded by a certain prediction context. I would encourage the authors to consider additional datasets characterized by both explanatory and contextual features.
> **Response:** We expanded the synthetic data generation process to incorporate two new scenarios: (1) a high-dimensional scenario with $p=1,000$ (Figure 1 of the PDF) and (2) a scenario where the contextual features are irrelevant (i.e., the lasso's home court, Figure 2 of the PDF). The first scenario is statistically challenging, though the contextual lasso offers highly competitive results. The second scenario is designed to evaluate the contextual lasso outside its home court. The contextual lasso remains competitive and recovers the ground truth for sufficiently large $n$. Regarding the real-data experiments, we highlight the third real-data example in Appendix G. We have also expanded the analysis of the house pricing data (see our response to Reviewer BSpt Comment 3), making for four real examples in total.
> 2. **Comment:** Although this work is aimed at capturing the prediction context by leveraging the expressive power of neural networks to map contextual features to a sparse coefficient vector, there are other works that take different approaches to the same end. For instance, Deleu and Bengio (2021) proposed adaptive optimizers for structured sparsity for deep learning, but also discusses unstructured sparsity () which is directly relevant to this work. There are also other works, such as Hallac, Leskovec, and Boyd (2015), that take a non-neural based approach to capturing prediction context but I believe are still relevant. Namely, Hallac, Leskovec, and Boyd (2015) addresses the house pricing problem (provided as an example in Section 1 of this work) using a generalization of the group lasso to a graph setting, where the graph represents the context defined by the distances between the house locations. Finally, there are also approaches that are not directly applicable to the setting considered in this paper, however, in a more general sense, they are relevant to inducing sparsity in neural networks and should be considered as a part of the related work  (e.g., Wang et al. 2020).
> **Response:** We have read all three papers and agree on their relevance. We will add each of them to the related works section. We briefly recap the differences in our work. Deleu and Bengio (2021) induce structured sparsity over network weights to obtain smaller, pruned networks that admit efficient computation. In our work, we leave the weights as dense and instead induce sparsity over the network's output for interpretability. Hallac, Leskovec, and Boyd (2015) introduce the Network Lasso with a different coefficient vector per observation and cluster these coefficients using a lasso-style regularizer. They consider problems similar to ours, for which contextual information is available, but do not induce sparsity over the coefficients (their regularizer clusters but does not sparsify).  Wang et al. (2020) study network quantization and devise a scheme for activation functions that output zeros and ones. Though our approach involves an activation that outputs zeros, we also allow a continuous output instead of a binary one. Moreover, their end goal differs from ours; whereas they pursue sparsity to reduce computational complexity, we pursue sparsity for interpretability.
> 3. **Comment:** As previously suggested, other sparse linear estimator variants have been proposed to utilize contextual information for the same or similar setting as the one considered in this work. Such models include Network Lasso (mentioned in the previous comment), Graphical Lasso and its associated extensions, among others. In that regard, I am wondering why the authors did not consider including such estimators as additional baselines. Moreover, since this work proposes a separate projection layer to induce sparsity in the output of a neural network, I also believe that a comparison to a conventional neural network with a dropout layer applied on its outputs ( values) would be appropriate.
> **Response:** As a new benchmark, we added the localized lasso  (Yamada et al. 2017). The localized lasso extends the network lasso by adding sparsity. We estimated the graph information from the contextual features using the nearest neighbour approach of  Yamada et al. (2017). A challenge with the localized lasso (and the network lasso) is that each new observation requires a new coefficient vector, i.e., additional optimization variables. Consequently, we found it difficult to scale the method to our main experiments (synthetic and real), so we ran smaller-scale experiments and plotted the results in Figure 3 of the PDF. The localized lasso's prediction loss improves with growing $n$, but at a slower rate than the contextual lasso. Regarding your last suggestion, it is an intriguing idea. However, initial experimentation indicates it does not work well in our setting. At $n=10,000$, $p=10$, and $m=2$, the dropout network yielded $\sim0.95$ relative test loss with the dropout rate set to reflect the true sparsity level (vs $\sim0.2$ for the contextual lasso). Cross-validating the dropout rate led to a dropout of zero. This result seems to be due to the contextually sparse setting involving a deterministic sparsity pattern, whereas dropout is random.

---

### Official Review · Reviewer_BSpt · 2023-07-07

**Soundness:** 3 good
**Presentation:** 4 excellent
**Contribution:** 3 good
**Rating:** 5
**Confidence:** 4

**Summary:**

In this paper, the authors present the contextual lasso, a new estimator that fits a sparse linear model on the explanatory features, allowing for the coefficients to vary as a function of the contextual features. This estimator generalizes the original lasso and the varying-coefficient model. The authors introduce a projection layer for efficient model training and furthermore, they develop a group version of the contextual lasso, which is analogous to the extension of lasso to the group lasso.

**Strengths:**

The paper is well-written and structured. The contextual lasso has the potential to provide substantial advantages for interpretation over neural networks, while leveraging the expressive power of these neural networks through the coefficient functions. The experiments conducted on property sales data (depicted in Figure 1), as well as real data tested in Section 4, are convincing to see the interpretability of the contextual lasso without compromising its predictive accuracy.

**Weaknesses:**

Although the method has not been explored in previous studies, it appears to be a relatively direct extension of the lasso and the varying-coefficient model. Besides the architecture of the projection layer for model training, the remaining parts of the paper, such as the extension to the group version, side constraints, pathwise optimization, and relaxed fit, are all straightforward extension from existing lasso literature.

The synthetic data experiments seem somewhat artificial. It's unclear whether these experiments provide any numerical evidence supporting the benefits of the contextual lasso. Given that the data is generated from the same model as the contextual lasso, it's expected that the method which leverages the true structure of the data would outperform or perform comparable to other methods. Furthermore, using the proportion of nonzero features as an interpretability metric is not appropriate, as interpretability is much broader than just sparsity. It would be more convincing to readers if the authors re-evaluate this experiment and supplement it with additional experiments using real data, for instance, in classification scenarios.


**Questions:**

Can the authors provide additional details regarding the experimental results on property sales shown in Figure 1? Merely displaying the fitted coefficients does not clearly indicate the reliability of the results. Are the model fits sufficiently accurate to validate that the interpretation of the coefficient functions is meaningful? Without further context of data, it's difficult to understand if the learned model accurately represents the true structure of the data.

Introducing the projection layer presents an issue with the non-differentiability of the objective function. How does this impact the actual training of the model? For example, is the learning process consistently stable across multiple random initializations?

In Section 4, the contextual lasso (or contextual group lasso) seems to outperform deep neural networks, but it's unclear whether the comparison is fair. The performance of deep neural networks can exceed the contextual lasso, depending on the chosen architecture. If the sample size is sufficient, deep neural networks should potentially offer superior performance compared to the contextual lasso, and depending on the applications, some practioners may favor the use of deep neural networks at the cost of sacrificing interpretability. It's important to make this clear in the section; currently, it reads that the contextual lasso surpasses deep neural networks in both predictive accuracy and interpretability.

In the first line of introduction, it might be more appropriate to revise the statement "Sparse linear models are a gold standard tool for interpretable machine learning," as interpretability really depends on the context of specific problems.

**Limitations:**

It would be beneficial to discuss the limitations of the contextual lasso, especially in relation to those of the ordinary lasso. For instance, the lasso often selects only one feature from a group of highly correlated features. Does the contextual lasso suffer a similar issue? Furthermore, how sensitive are the selected features to minor adjustment in the hyperparameter $\lambda$? Such stability considerations are important to enable better interpretation of the model.

---

> ### Author Rebuttal · Authors · 2023-08-10
>
> 1. **Comment:** Although the method has not been explored in previous studies...
> **Response:** Indeed, many of the techniques for the contextual lasso extend gracefully from the extensive, existing lasso literature. A crucial difference is that the existing techniques are designed for fixed coefficients. In contrast, we adapt all these techniques to the challenging setting of coefficients that vary in sparsity, magnitude, and sign.
> 2. **Comment:** The synthetic data experiments seem somewhat artificial...
> **Response:** The motivation for the existing synthetic design is to validate that the contextual lasso works correctly in the setting for which it is designed. However, we appreciate it is helpful to understand its properties in other settings. In response to your concern, we added a new set of experiments that evaluate the contextual lasso in a setting where the contextual features are irrelevant. Please see Figure 2 of the PDF. Notably, the contextual lasso is still able to recover the true nonzeros by learning a constant function for $\boldsymbol{\beta}(\mathbf{z})$, something it can achieve thanks to the bias term in each neuron. To address your question about interpretability metrics, we added a new experiment analyzing the stability of the selected features from the contextual lasso. Please see our response to your Comment 4. On your last point, we draw your attention to Appendix G of the supplement, which contains results for classification under real and synthetic data.
> 3. **Comment:** Can the authors provide additional details regarding the experimental results on property sales shown in Figure 1?...
> **Response:** We agree the paper would benefit from additional details regarding the housing data and will add a new section to the appendix with a complete explanation of the dataset. Regarding accuracy, we report below the test loss (over 10 random draws) from the contextual lasso and its competitors.
>
>    |                        | Relative loss    | Avg. sparsity    |
>    |------------------------|------------------|------------------|
>    | Deep neural network    | $0.515\pm0.003$  | $5.0\pm0.0$      |
>    | Contextual linear model| $0.505\pm0.002$  | $5.0\pm0.0$      |
>    | Lasso                  | $0.892\pm0.001$  | $5.0\pm0.0$      |
>    | Lassonet               | $0.521\pm0.003$  | $5.0\pm0.0$      |
>    | LLSPINN                | $0.729\pm0.031$  | $4.5\pm0.2$      |
>    | Contextual lasso       | $0.498\pm0.002$  | $2.9\pm0.3$      |
>
>    We will include these results in the appendix.
>
> 4. **Comment:** Introducing the projection layer presents an issue with the non-differentiability of the objective function...
> **Response:** The projection layer does involve some nondifferentiable operations (e.g., abs, sort). However, `Flux` provides gradients for these operations, e.g., sort is differentiated by permuting the gradients, and abs is differentiated by taking the subgradient zero at the point zero. We remark that the gradient obtained by differentiating through Algorithm 1 is the same as that obtained by using implicit differentiation with a convex solver (e.g., `cvxpylayers` of Agrawal et al. 2019), though the latter is slower. The training process is indeed fairly stable across random initializations. To demonstrate, we fit the contextual lasso under two random initializations and measured the hamming distance between the resulting sparsity patterns. We repeated this process under different values of $n$ and plotted the results in Figure 4 of the PDF. For $n>1000$, variation across the 50 explanatory features is relatively small.
> 5. **Comment:** In Section 4, the contextual lasso (or contextual group lasso) seems to outperform deep neural networks...
> **Response:** You are right that the deep neural network and contextual lasso should perform equally well (in terms of prediction) for a sufficiently large sample, and the former can outperform if the ground truth is nonlinear in the explanatory features. While we touch on this point in the experiments (see lines 237-238), we appreciate that it should be better drawn to the reader's attention. We will clearly state it in the paper's conclusion.
> 6. **Comment:** In the first line of introduction, it might be more appropriate to revise the statement "Sparse linear models are a gold standard tool for interpretable machine learning,"...
> **Response:** We agree that interpretability is context-specific, and sparsity/linearity alone may not be sufficient for interpretability in some cases. We will rewrite the sentence as: "Sparse linear models are one of several core tools in interpretable machine learning, ...".
> 7. **Comment:** It would be beneficial to discuss the limitations of the contextual lasso...
> **Response:** We will add the following paragraph to a new limitations section in the appendix:
> > The contextual lasso has strong prediction, selection, and interpretation properties but also has several weaknesses. As with the ordinary lasso, the contextual lasso tends to select a single feature from a group of highly correlated features. This shortcoming could be remedied by introducing an $\ell_2^2$-penalty in a spirit similar to the elastic net (Zou and Hastie 2005). Another potential drawback is that the contextual lasso does not guarantee the complete exclusion of an explanatory feature. That is, it cannot ensure the coefficient for a feature is nonzero for every possible $\mathbf{z}$, which might limit interpretability in certain settings. Full exclusion could be achieved by adding an $\ell_1$-penalty on the weights of the first layer, similar to the lassonet. Finally, as is typical of neural networks, the objective function of the contextual lasso is nonconvex. Nonconvexity complicates the analysis of its computational and statistical properties.
>
>    Regarding stability, it is our experience that the solution is not sensitive to small changes in $\lambda$ due to continuity of the regularizer.

---

### Official Review · Reviewer_Eybr · 2023-07-08

**Soundness:** 4 excellent
**Presentation:** 4 excellent
**Contribution:** 3 good
**Rating:** 7
**Confidence:** 3

**Summary:**

This paper proposes a contextual lasso, which makes the coefficients of features sparse depending on the context.
Unlike the standard lasso, the contextual lasso regularizes prediction loss with the expectation of L1 regularization.
The sparse coefficients are produced by neural networks whose last layer is a projection layer that maps dense coefficients onto l1-ball.
The experimental results on synthetic and real datasets show that the contextual lasso outperforms the linear lasso and 'dense' contextual lasso while achieving the high sparseness of the coefficients.



**Strengths:**

Overall, this paper is well-organized and well-written. Section 2, which describes the proposed method, is easy-to-follow.
The proposed contextual lasso is technically sound, and the high expandability similar to the lasso is described.
The experiments are organized well, and the experimental results are convincing.


**Weaknesses:**

In the experiments, the contextual lasso is assessed only on tabular datasets with relatively low dimensional features.
Therefore, it is unclear whether the contextual lasso consistently works well for high-dimensional features and data other than tabular data, such as images and texts.


**Questions:**

- When contextual features are useful but explanatory features are uninformative, I think that the predictions using the contextual lasso would be worse than the standard neural networks, which use both contextual and explanatory features as input. Is there any solution to this?
- It would be easier to read this paper if figures were displayed at the top of the page.

**Limitations:**

I could not find the description of the limitations.

---

> ### Author Rebuttal · Authors · 2023-08-10
>
> 1. **Comment:** In the experiments, the contextual lasso is assessed only on tabular datasets with relatively low dimensional features. Therefore, it is unclear whether the contextual lasso consistently works well for high-dimensional features and data other than tabular data, such as images and texts.
> **Response:** We focus on tabular datasets as these are the most common use case for the lasso and its cousins. We will clarify the introduction to state our focus. The question of higher-dimensional datasets is interesting, and we now include experiments where the number of explanatory features is 1000 ($20\times$ larger than before), and 990 of these are zero for all values of $\mathbf{z}$. The results are plotted in Figure 1 of the PDF. Importantly, the contextual lasso can still filter out the irrelevant explanatory features and achieves near-perfect support recovery for large $n$.
> 2. **Comment:** When contextual features are useful but explanatory features are uninformative, I think that the predictions using the contextual lasso would be worse than the standard neural networks, which use both contextual and explanatory features as input. Is there any solution to this?
> **Response:** We expect the contextual lasso to outperform a standard neural network in this setting due to its ability to remove the effect of the explanatory features by zeroing out their coefficients. In contrast, a standard neural network has no such capability, i.e., the effect of the explanatory features will flow through the network unencumbered.
>
> 3. **Comment:** It would be easier to read this paper if figures were displayed at the top of the page.
> **Response:** We will make this change.
>
> 4. **Comment:** I could not find the description of the limitations.
> **Response:** We have added a dedicated discussion on limitations---please see our response to Comment 7 of Reviewer BSpt.

---

> > ### Comment · Reviewer_Eybr · 2023-08-14
> > **Thank you**
> >
> > Thank you for your response and for performing the additional experiment.
> > The experiment has convinced me that the proposed method is likely to work well even for high-dimensional data.
> > As I think your paper deserves to be accepted, I will keep my score.

---

### Author Rebuttal · Authors · 2023-08-10

We thank all the reviewers for their time and careful reviews of our paper. We have now responded to all the questions and concerns raised. Besides making clarifications and textual modifications to the manuscript, we made the following major improvements:
- High-dimensional experiments with $p=1000$ explanatory features. For Reviewers Eybr and mL1j.
- Experiments where the contextual features are irrelevant. For Reviewers BSpt and mL1j.
- New benchmark comparator: the localized lasso. For Reviewers mL1j and kkJW.
- Stability analysis of the contextual lasso. For Reviewer BSpt.
- Time complexity analysis of the contextual lasso. For Reviewer kkJW.
- Extended analysis of the housing price dataset of the introduction. For Reviewer BSpt.
- Dedicated discussion of the contextual lasso's limitations. For Reviewers Eybr and BSpt.
- Discussion on the partition of the contextual/explanatory features. For Reviewers DGcD and kkJW.

If the reviewers have further questions, we are happy to address them.

**New figures are available in the attached PDF.**

*References*

1. Agrawal, A., Amos, B., Barratt, S., Boyd, S., Diamond, S., and Kolter, J. Z. (2019). "Differentiable convex optimization layers". Advances in Neural Information Processing Systems. Vol. 32.

2. Bunea, F., Tsybakov, A. B., and Wegkamp, M. H. (2007). "Aggregation for Gaussian regression". Annals of Statistics 35.4, pp. 1674–1697.

3. Deleu, T. and Bengio, Y. (2021). "Structured sparsity inducing adaptive optimizers for deep learning". arXiv: 1711.07592.

4. Duchi, J., Shalev-Shwartz, S., Singer, Y., and Chandra, T. (2008). "Efficient projections onto the ℓ1-ball for learning in high dimensions". Proceedings of the 25th International Conference on Machine Learning, pp. 272–279.

5. Hallac, D., Leskovec, J., and Boyd, S. (2015). Network lasso: Clustering and optimization in large graphs. Association for Computing Machinery, pp. 387–396.

6. Lemhadri, I., Ruan, F., Abraham, L., and Tibshirani, R. (2021). "LassoNet: A neural network with feature sparsity". Journal of Machine Learning Research 22, pp. 1–29.

7. Raskutti, G., Wainwright, M. J., and Yu, B. (2011). "Minimax rates of estimation for high-dimensional linear regression over ℓq -balls". IEEE Transactions on Information Theory 57.10, pp. 6976–6994.

8. Schmidt-Hieber, J. (2020). "Nonparametric regression using deep neural networks with ReLU activation function". Annals of Statistics 48 (4), pp. 1875–1897.

9. Al-Shedivat, M., Dubey, A., and Xing, E. (2020). "Contextual explanation networks". Journal of Machine Learning Research 21, pp. 1–44.

10. Shen, X., Pan, W., Zhu, Y., and Zhou, H. (2013). "On constrained and regularized high-dimensional regression". Annals of the Institute of Statistical Mathematics 65.5, pp. 807–832.

11. Wang, P., He, X., Li, G., Zhao, T., and Cheng, J. (2020). "Sparsity-inducing binarized neural networks". Vol. 34, pp. 12192–12199.

12. Yamada, M., Takeuchi, K., Iwata, T., Shawe-Taylor, J., and Kaski, S. (2017). "Localized lasso for high-dimensional regression". Proceedings of the 20th International Conference on Artificial Intelligence and Statistics 54, pp. 325–333.

13. Zhang, Y., Wainwright, M. J., and Jordan, M. I. (2014). "Lower bounds on the performance of polynomial-time algorithms for sparse linear regression". Proceedings of the 27th Conference on Learning Theory. Vol. 35, pp. 921–948.

14. Zou, H. and Hastie, T. (2005). "Regularization and variable selection via the elastic net". Journal of the Royal Statistical Society: Series B (Statistical Methodology) 67 (2), pp. 301–320.

---

> ### Comment · Area_Chair_2W8E · 2023-08-17
> **Acknowledgement of rebuttal**
>
> Dear authors, thank you very much for your detailed rebuttal (which we are currently discussing).
>  Best, AC

---

### Decision · Program_Chairs · 2023-09-21

**Decision:**

Accept (poster)

**Comment:**

There are two "borderline" reviews which mention several strong points, such as the novelty of the proposed integration of explanatory and contextual information, the idea  using a sparsity-inducing projection layer, and the experimental evaluation.
On the negative side, these reviewers mention problems related to the lacking formal definition of "context", the limited theoretical depth and the conceptual similarity to other approaches.

The more "positive" reviewers also mention the novelty of the idea and quality of the experimental evaluation, which, for them, outweigh some weaknesses, such as possible limitations due to the assumption of an existing  "explanatory\contextual structure" in the underlying problem domain.

After going again over the paper, all reviews, the rebuttal and the discussion, I come to the conclusion that this is indeed a borderline case, but for me the strengths regarding the over-all novelty of the idea and the conclusive experimental evaluation (slightly) outweigh several conceptual weaknesses. However, since this is not a clear-cut case, I would like to encourage the authors to improve the final version of their paper by including further clarifications and experimental results from the discussion phase.